# Hybrid-DIA: intelligent data acquisition integrates targeted and discovery proteomics to analyze phospho-signaling in single spheroids

Ana Martínez-Val [1], Kyle Fort [2], Claire Koenig [1], Leander Van der Hoeven [1], Giulia Franciosa [1], Thomas Moehring [2], Yasushi Ishihama [3], Yu-ju Chen [4], Alexander Makarov [2], Yue Xuan [2] ✉ & Jesper V. Olsen [1] ✉

Achieving sufficient coverage of regulatory phosphorylation sites by mass spectrometry (MS)-based phosphoproteomics for signaling pathway reconstitution is challenging, especially when analyzing tiny sample amounts. To address this, we present a hybrid data-independent acquisition (DIA) strategy (hybrid-DIA) that combines targeted and discovery proteomics through an Application Programming Interface (API) to dynamically intercalate DIA scans with accurate triggering of multiplexed tandem mass spectrometry (MSx) scans of predefined (phospho)peptide targets. By spiking-in heavy stable isotope labeled phosphopeptide standards covering seven major signaling pathways, we benchmark hybrid-DIA against state-of-the-art targeted MS methods (i.e., SureQuant) using EGF-stimulated HeLa cells and find the quantitative accuracy and sensitivity to be comparable while hybrid-DIA also profiles the global phosphoproteome. To demonstrate the robustness, sensitivity, and biomedical potential of hybrid-DIA, we profile chemotherapeutic agents in single colon carcinoma multicellular spheroids and evaluate the phospho-signaling difference of cancer cells in 2D vs 3D culture.

Liquid chromatography-tandem mass spectrometry (LC-MS/MS) acquisition strategies for proteomics can be divided into two main categories: discovery and targeted proteomics methods. The aim of discovery approaches is to achieve the most comprehensive coverage of the proteome or sub-proteome under investigation. Although still far from completeness, the most representative MS acquisition method to achieve this in single-shot analysis is data-independent acquisition (DIA)[1]. DIA has proven capable of maximizing the number of identifications obtained per sample, especially when studying post-translational modification (PTM) landscapes[2–4]. For example, DIA-based discovery proteomics is a powerful technology for studying global changes in the phosphoproteome[5,6] in cells, tissues, and organisms. Implementation of spectral library-free directDIA analysis in the context of phosphoproteomics has emerged as a powerful method to increase the depth and facilitate high-throughput single-shot phosphoproteomics analysis by removing the requirement of off-line library generation[5]. The study of the phosphoproteome implies an extra layer of complexity in MS analysis, since the diversity of phosphorylated peptide species is higher but their abundance is generally lower than that of unmodified peptides. Therefore, phosphoproteomics typically requires enrichment of phosphopeptides prior to MS analysis, limiting the scope of the analysis when sample availability is scarce. Fortunately, the phosphoproteomics technology has advanced significantly in recent years in terms of sensitivity and robustness as

[1]Novo Nordisk Foundation Center for Protein Research, University of Copenhagen, Copenhagen, Denmark. [2]Thermo Fisher Scientific, Bremen, Germany. [3]Kyoto University, Kyoto, Japan. [4]Academia Sinica, Taipei, Taiwan. ✉e-mail: yue.xuan@thermofisher.com; jesper.olsen@cpr.ku.dk

optimal amounts required for phosphopeptide-enrichment prior to MS analysis has been reduced by a factor of ten from ~2 mg to 200 μg of peptide input[7–9]. In a recent study, Ochoa et al reanalyzed thousands of published phosphoproteomics experiments and created a human reference phosphoproteome comprised of more than 119,000 different phosphorylated species[10]. Contrastingly, single-shot analysis of phosphoproteomes, the most popular strategy for high-throughput analyses, can only provide partial coverage of the phosphorylated proteome. This makes the biological interpretation challenging especially when sample amounts are limited. Nonetheless, even with the latest advancements in library-free DIA and phosphoproteomics[5,6], single-shot phosphoproteome depth is not close to completeness, and many functional phosphosites of interest might not be detected due lack of sensitivity or high complexity of the sample. These limitations become more evident when sample amount is limited, for example in clinical specimens, such as formalin-fixed paraffin-embedded (FFPE) samples[11] or single organoids. Yet, despite the latest boost in depth achieved by single-shot phosphoproteomics due to DIA, many biologically important phosphopeptide targets of low abundance are often missed in phosphoproteomics experiments. This shortcoming makes certain biological systems inaccessible to traditional phosphoproteomics analysis due to their inherent limited material, such as analysis of phosphoproteomes from single spheroids or organoids, tumor fine needle aspiration biopsies, or even single-cells. In these scenarios, the protein amount available for phosphoproteomics analysis is suboptimal, and in the best-case scenario, only allows for single-injection LC-MS/MS analysis. Therefore, for restricted biological matrices it is essential to enhance sensitivity of the analysis to maximize the phosphoproteome coverage in each MS run.

In distinction to discovery proteomics, targeted proteomics approaches provide improved detection and quantification of a pre-defined set of peptides with good accuracy and precision across multiple runs; however, single and parallel reaction monitoring (SRM/PRM) methods require extensive method optimization that, among other factors, limits the number of target peptides that can be accurately monitored. To address this limitation, intelligent acquisition methods have been developed, such as spike-in triggered PRM acquisition methods (i.e., SureQuant[12], TOMAHAQ[13], Pseudo-PRM[14]) in which targeted scans are triggered by detection of synthetic heavy-labeled peptides spiked into the samples before MS analysis. However, while the existing intelligent MS data acquisition methods improve the sensitivity and reproducibility of phosphoproteomics, especially by ensuring accurate quantification of key phosphorylation pathway markers, they are restricted in coverage to a limited set of predefined peptide targets and will be missing the global (phospho)proteome footprint.

Translational scientists face a dilemma when having to choose between comprehensive discovery proteomics-based profiling and sensitive targeted quantitation[15], especially when analyzing large sample cohorts. Discovery proteomics is commonly used for biomarker identification, having a great potential for unveiling prognostic and predictive biomarkers. However, it still lacks the sensitivity to accurately quantify all the biomarkers of interest. Therefore, in the validation phase, targeted MS quantitation of the potential markers usually have to be employed. This leads to high cost, time loss and additional sample consumption.

To address these challenges, we develop an intelligent MS data acquisition strategy termed hybrid-DIA, that combines comprehensive proteome profiling via data-independent-acquisition mass spectrometry with simultaneous on-the-fly triggering of parallel reaction monitoring (PRM) and multiplexed MS/MS (MSx) scans for sensitive and accurate quantification of the predefined marker peptides. This hybrid-DIA MS acquisition strategy substantially increases throughput and coverage while reducing sample consumption. It introduces the ability to combine unbiased DIA-based profiling with hypothesis-driven MS acquisition approaches in one run. The hybrid-DIA acquisition strategy uses an Application Programming Interface (API) to dynamically intercalate DIA scans with multiplexed tandem MS scans of predefined (phospho)peptide targets by spiking-in heavy stable isotope-labeled (phospho)peptide standards. In this work, we benchmark hybrid-DIA to show its benefits when compared to conventional DIA and triggered targeted proteomics acquisition methods.

Hybrid-DIA method acquisition can be employed in any application where accurate and sensitive quantification of pre-defined targets needs to be combined with discovery-based proteomics analysis in a single-shot strategy. In this project, as a specific showcase, we demonstrate that hybrid-DIA is an excellent strategy for high-sensitivity phosphoproteomics, by showing how it maximizes the information retrievable from challenging low-level phosphoproteomics samples. In such cases, the minute sample amount retrieved after phosphopeptide-enrichment constrains the MS analysis and forces a decision between discovery analysis (i.e., DIA) or targeted quantification of key signaling pathways (i.e., SureQuant). In this work, we demonstrate how hybrid-DIA can maximize the information obtained from low-input phosphoproteomics in one single analysis, improving the surveying of specific signaling pathways due to the targeted analysis of key phospho-sites coupled to the coverage achieved by DIA analysis, which is needed for downstream discovery-based pathway reconstruction. Furthermore, we showcase the applicability of hybrid-DIA-based high-sensitivity phosphoproteomics to dissect drug actions by describing the mechanism of action of the chemotherapeutic drug 5-fluorouracil (5-FU) in single colon cancer multicellular spheroids compared to conventional monolayer cell culture.

## Results
### Implementation of an API to enable targeted and discovery proteomics

Hybrid-DIA is an intelligent MS data acquisition strategy implemented through an Application Programming Interface (API) in the Tune software controlling an Orbitrap™ Exploris™ 480 mass spectrometer[16] (see Online Methods and Supplementary Note 1). The acquisition method combines a standard DIA acquisition scheme consisting of a full MS scan followed by a flexible number of MS/MS precursor isolation windows with on-the-fly triggering of intercalated multiplexed MS/MS (MSx) scans. In an MSx scan, fragments from multiple precursors isolated are stored together in the HCD cell, and sent to the Orbitrap for detection as a single MS/MS scan. In hybrid-DIA, MSx scans consist of spiked-in heavy stable isotopically labeled peptide standards (IS) and their predicted endogenous (ENDO) counterparts based on a predefined precursor inclusion list (Fig. 1a). The peptide precursor list needs to be generated in advance by performing PRM analysis of the heavy stable isotopically labeled peptide standards that will be spiked into the biological sample of interest. From such an analysis, the chromatographic retention time and the top N fragment ions will be selected and used to generate the inclusion list that will be used by the API (Supplementary Note 1). During a full MS scan, if any of the IS peptides on the inclusion list for a given time window is detected, the API will trigger a MS/MS scan with the observed IS peptides (Fig. 1a, Supplementary Fig. 1A). Automatic matching of a minimum number of predefined fragment ions with high mass accuracy in the MS/MS of the heavy isotope-labeled standards triggers additional multiplexed MS/MS spectra (MSx) of each of the individual heavy IS co-analyzed with their corresponding endogenous peptide (ENDO), respectively (Supplementary Fig. 1B, C). The triggered MSx scans of the IS and ENDO peptides (IS/ENDO-MSx) are acquired with narrow quadrupole isolation window and differential ion injection times (IT) to equalize the precursor abundances and thereby augment quantitative accuracy and sensitivity (Fig. 1b). Using the hybrid-DIA API, all of the triggered targeted PRM and MSx scans are performed in the same scan

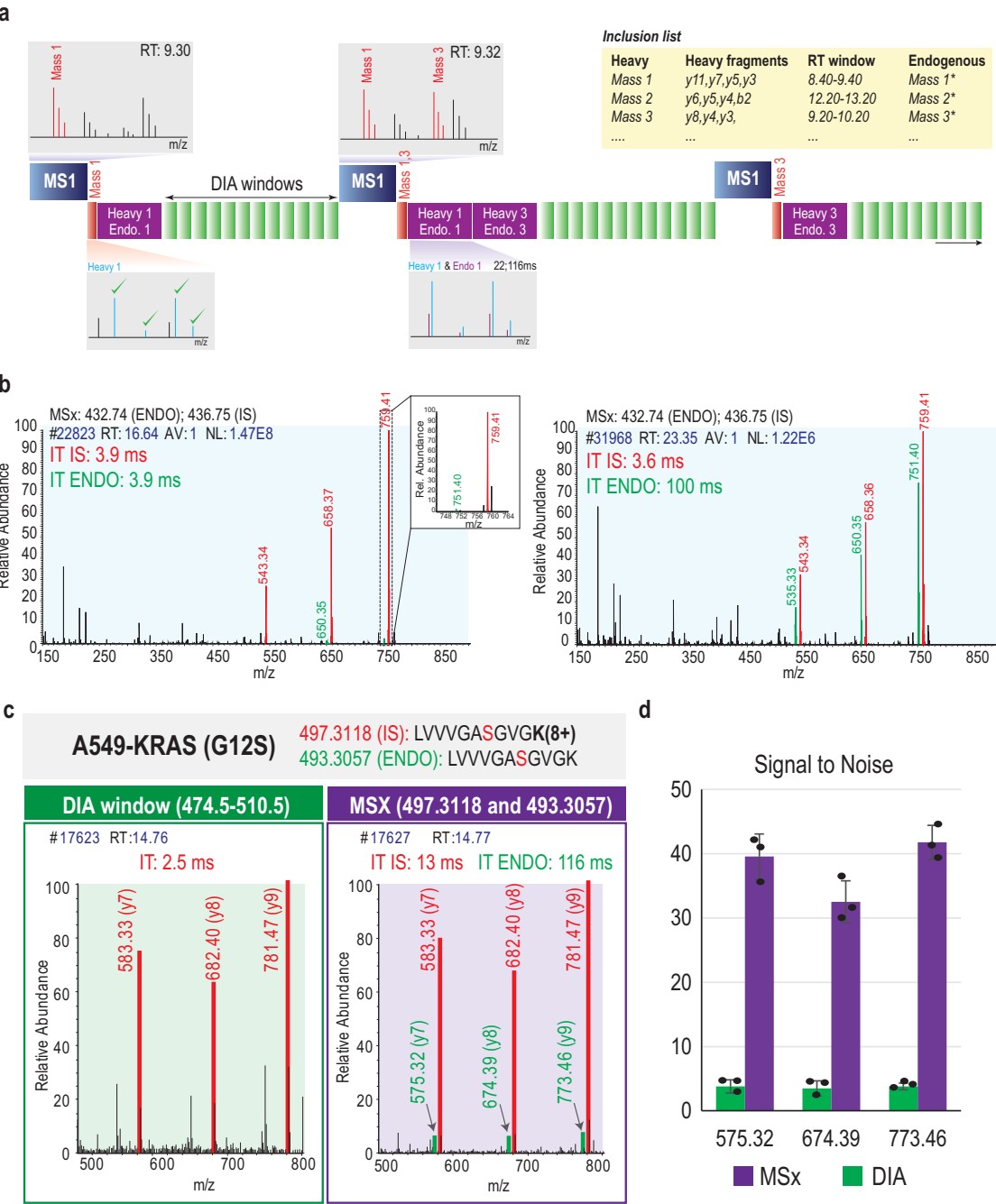

**Fig. 1 | Description of hybrid-DIA method. a** Description of the hybrid-DIA acquisition method. The figure represents a scheme of three acquisition cycles. A hybrid-DIA acquisition scan cycle starts with a full-scan or MS1 scan that it is acquired at high resolution. The hybrid-DIA API scans the MS1 for the precursor m/z values in the inclusion list. If any is found (e.g., Mass 1) a targeted scan of this IS precursor m/z will be triggered as part of the subsequent MS/MS scan cycle (red box). The API will on-the-fly match the obtained fragments from that PRM scan (red box) to the IS peptide fragment values in the inclusion list. If the fragments match the expected values, then a targeted multiplexed scan (MSx) will be added to the cycle (purple box). This MSx scan (purple box) will simultaneously analyze the heavy (IS) and the endogenous (ENDO) peptide using non-synchronous injection times. Subsequently, the predefined DIA windows of the MS method will be acquired. Once the complete scan cycle is done, a new one will start and the process will be repeated in each cycle when heavy peptides are found in the MS1 scan. The box width reflects the injection times used for each scan, but it is not set to scale. **b** Example of the increase in sensitivity by using differential injection time in the multiplexed (MSx) scans. In red, spiked-in heavy stable isotopically labeled peptide standard (IS), and in green the triggered endogenous peptide (ENDO). **c** Example of a DIA scan and an IS/ENDO MSx scan from the same cycle where the heavy peptide carrying the G12S mutation, as well as its endogenous counterpart were isolated and fragmented. The top 3 y-fragments of each peptide are highlighted, in red the heavy fragments and in green the endogenous fragments. **d** Signal-to-noise (S/N) measured in the top 3 y-fragments in the endogenous G12S peptide in a DIA window (green) and in the MSx IS/ENDO scan (purple) in the same acquisition cycle. Height of the bars is the average S/N of three replicates, error bars are the standard deviation. RT: retention time; IT: injection time; m/z: mass to charge. Source data are provided as a Source Data file.

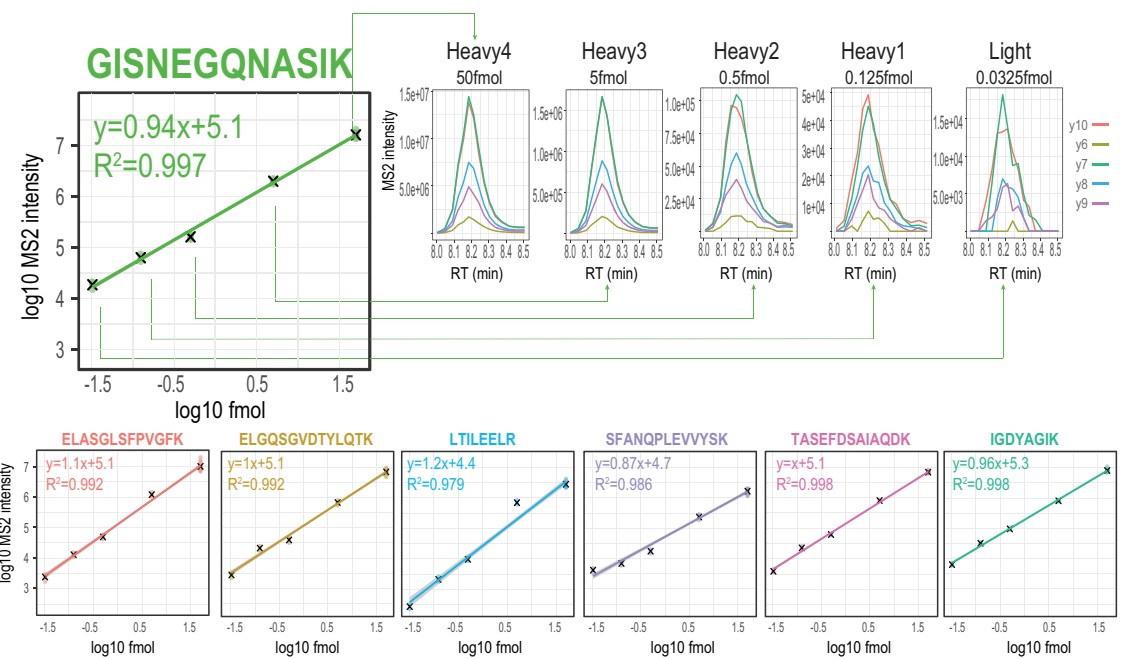

**Fig. 2 | Evaluation of quality of targeted data generated using hybrid-DIA.**
Demonstration on the accuracy, precision, and linearity of the quantification in the MSx scans using the Pierce Suitability Standard Mix, which is comprised of 7 peptides, each one with 5 isotopologue sequences present in a dilution series ranging from 0.5 pmol/μl to 0.3 fmol/μl. The isotopologue with the highest concentration (Heavy4) was used as a triggering peptide, and the subsequent peptides were used as endogenous counterparts, and were triggered sequentially in the acquisition cycle. RT: retention time. Source data are provided as a Source Data file.

cycle as the DIA windows (Fig. 1a). As a result, hybrid-DIA raw files contain both unbiasedly acquired DIA data as well as a selection of targeted scans of peptides with higher sensitivity and better quantitative accuracy and precision. To process hybrid-DIA raw files, it is necessary to separate the DIA scans from the IS/ENDO-MSx scans. For retrieving the DIA data, we employed the HTRMS convertor tool co-installed with Spectronaut software, which was used to analyze the DIA data. Moreover, we have developed an in-house analysis pipeline to extract the IS/ENDO-MSx scans and quantify all IS/ENDO peptide pairs detected. To do this, we first extract all IS/ENDO-MSx scans into a separate mzML[17] file, whilst simultaneously extracting the information on the differential IT used in the MSx scans. Next, we load the mzML file into Skyline[18] to readout the raw fragment ion intensities from each pair of IS/ENDO peptides. Finally, the resulting files are loaded into an R-shiny app that we have designed to correct for the differential injection time (Supplementary Fig. 2A, B, Supplementary Note 2), and determine the area-under-the-curve (AUC) of both ENDO and IS peptides. Additionally, the R-shiny app enables visual inspection of the MSx scans and the resulting quantification, scaled by conditions or as ENDO/IS ratios (Supplementary Fig. 2A, B, Supplementary Note 2). Alternatively, the processing of the targeted scans can be performed directly from the raw files using the SpectroDive (v11.1) software suite for analysis of targeted proteomics data (Supplementary Fig. 2C).

Hybrid-DIA is designed to boost the sensitivity in detection and quantitation of the endogenous peptides in the MSx IS/ENDO scans. This is achieved by using differential ion injection times to maximize fill times for the ENDO peptide, which is typically of lower abundance than the spiked-in IS peptide. As an example of this gain in quantitative performance by hybrid-DIA, we analyzed the A549 lung adenocarcinoma cells (ATCC: CCL-185), which harbors an oncogenic G12S mutation in KRAS, where glycine is mutated to serine[19]. We spiked in 25 fmol of a synthetic heavy stable isotopically labeled peptide corresponding the mutated peptide into 100 ng A549 peptide extracts and analyzed the sample using hybrid-DIA. When comparing the signals of the endogenous and heavy peptides in the DIA isolation window against the consecutive MSx IS/ENDO scan, we evidently see an improved

detection of the endogenous peptides, which accounts for an 8-fold gain in signal-to-noise (Fig. 1c, d).

Furthermore, to assess the accuracy, precision, and linearity of the quantification from the MSx scans derived from the hybrid-DIA files, we used the Pierce LC-MS/MS System Suitability Standard mixture comprising 7 different peptides, each one with 5 isotopologue sequences[20] present in a dilution series covering more than three orders of magnitude in dynamic range ranging from 0.5 pmol/μl to 0.3 fmol/μl. We injected 0.1 μl of the Pierce Suitability Standard mixture, and used the more abundant isotopologue (50 fmol on column) to trigger MSx scans of the remaining 4 isoforms (5, 0.5, 0.125 and 0.0325 fmol, respectively). Samples were analyzed using the Whisper flow technology 40 samples per day (SPD) gradient on an Evosep One LC system coupled to an Orbitrap Exploris 480 mass spectrometer operating with the hybrid-DIA API. Specifically, we acquired MSx scans using a maximum of 116 ms injection time and automatic gain control (AGC) with target value of 1e6. As a result, all seven peptides were correctly detected and all four isotopologues quantified for each peptide. Using our hybrid-DIA analysis pipeline (Supplementary Note 2), we quantified the intensities from each of the isotopologues measured in the hybrid-DIA scan. We found that the targeted MSx scans allowed to correctly quantifying amounts as low as 0.0325 fmol, and the quantification showed perfect linearity for the entire dynamic range covered (Fig. 2).

## Hybrid-DIA improves the limit of detection and quantification of predefined targets in phosphoproteomics whilst preserving the coverage of the phosphoproteome

The spectral library-free directDIA MS analysis strategy has recently emerged as a high-throughput and straightforward approach for discovery-based phosphoproteomics[5]. However, such single-shot phosphoproteomes are still limited in coverage and are far from completeness[10], and many phosphopeptides of interest might not be detected and quantified properly. Moreover, site-specific phosphorylation is a dynamic sub-stoichiometric post-translational modification (PTM) requiring specific phosphopeptide enrichment prior to MS

analysis, which makes sample amounts available critical for effective phosphoproteomics analysis. Importantly, many sample types of biomedical interest are limited in protein amount (e.g., FACS sorted cells, fine needle aspiration biopsies, FFPE samples or single spheroids) restricting the possibility to perform both discovery proteomics and targeted MS validation from the same material. The hybrid-DIA methodology can alleviate this dilemma of choosing between DIA or PRM analysis, and thereby maximize the knowledge derived from a single sample, which is of special relevance for high-sensitivity phosphoproteomics applications.

To demonstrate the benefits of hybrid-DIA in terms of improved sensitivity, we benchmarked it against conventional DIA analysis in a human cancer cell line model for sensitive phosphoproteome analysis. Using A549 human lung adenocarcinoma cells, we performed phospho-enrichment from decreasing amounts of tryptic peptide digests, starting from 30 μg and down to 2.5 μg of peptide input. Samples were prepared in quadruplicates, each phosphopeptide-enrichment performed independently (Fig. 3a). To assess the potential of hybrid-DIA for measuring a predefined panel of phosphopeptides, we used the commercially-available SureQuant™ Multipathway Phosphopeptide Standard mixture containing 131 heavy stable isotope labeled tryptic phosphopeptides of relevance covering seven major cellular signaling pathways. 50 fmol of the mixture was added to all samples, and subsequently half of them were analyzed in DIA mode, and the other half using the hybrid-DIA approach (Fig. 3a). Hybrid-DIA files were processed using the pipeline described in Supplementary Note 2. Importantly, the comparison with the results obtained from SpectroDive anlaysis of the same dataset shows that the results from both approaches are highly comparable (Supplementary Fig. 2D).

The phosphopeptides contained in the SureQuant™ Multipathway Phosphopeptide Standard mixture are evenly distributed across the 20SPD chromatographic gradient (Supplementary Fig. 3A), and their endogenous counterparts are very diverse in MS signal intensities spanning several orders of magnitude. To prove the improved limit of detection of hybrid-DIA in the MSx scans, we extracted the ion chromatograms (XICs) of three peptides from the panel with different abundances: AKT1S1:T246 (high abundance), TSC2:S939 (medium abundance) and PLCG1:Y783 (low abundance). Whilst the phosphopeptide of high abundance (AKT1S1:T246) is clearly detected both in the MSx scans in hybrid-DIA mode and in the MS/MS scans in standard DIA mode, it is clear that for phosphopeptides of lower abundance (TSC2:S939 and PLCG1:Y783), the retrieved signal for both is lower or missing in standard DIA, but readily detected in MSx scans in hybrid-DIA, even at input amounts as low as 2.5 μg prior to phospho-enrichment (Fig. 3b, c). When evaluating the sensitivity of the whole panel of targeted phosphopeptides, the lower sensitivity and limit of detection of DIA is especially evident for lower abundant peptides, whilst for highly abundant phosphopeptides the quantitative performance of DIA is comparable to that of hybrid-DIA MSx IS/ENDO scans (Fig. 3c). Importantly, the quality of the precursor-to-fragment ion pair transitions measured in MSx IS/ENDO scans from hybrid-DIA is superior to that of standard-DIA MS/MS scans (Fig. 3d). We assessed the quality of the transitions based on two targeted MS quality measures provided by Skyline: peptide-peak found ratio (PPFR) and dot product ratio (DOTPR)[18]. PPFR indicates the proportion of transitions in which Skyline determines there is a peak co-eluting with the primary peak, whilst DOTPR measures whether the transition peak areas in the two label types are in the same ratio of each other. When analyzing both datasets in Skyline, we found that all of the IS/ENDO peptide pairs detected in hybrid-DIA shows a PPFR above 0.5, whilst less than 50% of the ones found in the standard-DIA dataset are above that threshold. When filtering by DOTPR, the peptides in the hybrid-DIA dataset are reduced proportionally to the input amount, but still retaining more than 50% of the total found peptides, whereas the number of targeted peptides from the DIA part that can be used for downstream analysis

and quantification is less than 25% of the total (Fig. 3d). This demonstrates that hybrid-DIA provides better coverage the targeted endogenous peptides based on the MSx scans triggered by the spiked-in IS peptide panel.

Moreover, we also evaluated the quality of the quantification obtained from targeting the endogenous peptides measured by the ratio of endogenous–to-heavy spiked-in peptide. We already showed (Fig. 3c, d) that hybrid-DIA provides more depth in the targeted panel, but also the precision of the measurements between replicates in higher than those in standard-DIA (Fig. 3e).

Finally, to assess if the improved quantitative precision derived from the improved MS-signal in MSx IS/ENDO scans also impacts the quantitative accuracy, we compared the ratios for each input amount versus the 30 μg input samples (i.e., 30 μg vs 30 μg, 30 μg vs 20 μg, 30 μg vs 10 μg, 30 μg vs 5 μg, 30 μg vs 2.5 μg). We used the quantification of the DIA-based full phosphoproteome in both datasets (standard and hybrid-DIA) as the reference, and compared it to the quantification obtained for the panel of targeted peptides as the measured ratios are expected to be constant when comparing different input amount pairwise.

For the comparisons, we used the IS/ENDO ratio as the quantification value to calculate the ratios for the panel of targeted peptides. Contrarily, we employed the log2-transformed MS2 intensities for all peptides detected by Spectronaut from the DIA windows to calculate ratios for the standard DIA. As expected, when comparing two replicates of 30 μg input, the calculated log2 fold change ratio is very close to zero for both the full phospho-proteome and the targeted peptides, in both standard-DIA and hybrid-DIA (Fig. 3f). However, for the all other comparisons, the ratio increased as expected, but the median ratios measured from the targeted phospho-peptides and the rest of the phospho-peptides deviated more in standard-DIA than in hybrid-DIA, especially for the largest ratio, in which the lowest amount (i.e., 2.5 μg) was used (Fig. 3f). This analysis demonstrates that the hybrid-DIA approach provides both better quantitative accuracy and precision for the targeted set of peptides compared to conventional DIA.

One important concern that might arise when comparing standard DIA against hybrid-DIA is whether or not the inclusion of targeted scans during a normal DIA run affects the scan cycle time significantly, and if so, how it impacts the overall identification and quantification of peptides. To assess this, we calculated the percentage of measurement time employed by hybrid-DIA PRM and MSx scans in the A549 phosphoproteome experiment described above and found that almost 20% of the MS/MS scans are triggered by the API, comprising one third of the total MS/MS acquisition time not considering the full scans (Fig. 4a). However, when comparing the number of phosphopeptides identified by directDIA+ using Spectronaut (v17) in each method, we did not see any notable difference between conventional DIA and hybrid-DIA runs (Fig. 4b). However, to evaluate in more depth the relationship between the total number of IS/ENDO targets on the inclusion list and the impact on DIA performance, we carried on an experiment targeting an increasing number of targeted phosphopeptides (50, 75 and 100 targeted peptides) (Fig. 4c, Supplementary Figure 3A) with hybrid-DIA while decreasing the LC gradient to 40 SPD. As expected, an increase in the fraction of cycle time devoted to hybrid-DIA scans is observed as more targets are added to the inclusion list (Fig. 4d). This is accompanied by a slight decrease in identifications, which is proportional to the number of estimated targets per LC gradient minute in the hybrid-DIA method (Fig. 4e, f). This data could be used to predict the maximum number of peptides that are realistic to target in a given gradient length by extrapolation with for example an expected loss of 25% in the number of DIA identifications when targeting 8 peptides per LC gradient minute (Fig. 4f).

Finally, we evaluated the reproducibility between the quantification obtained from the untargeted directDIA search of the standard

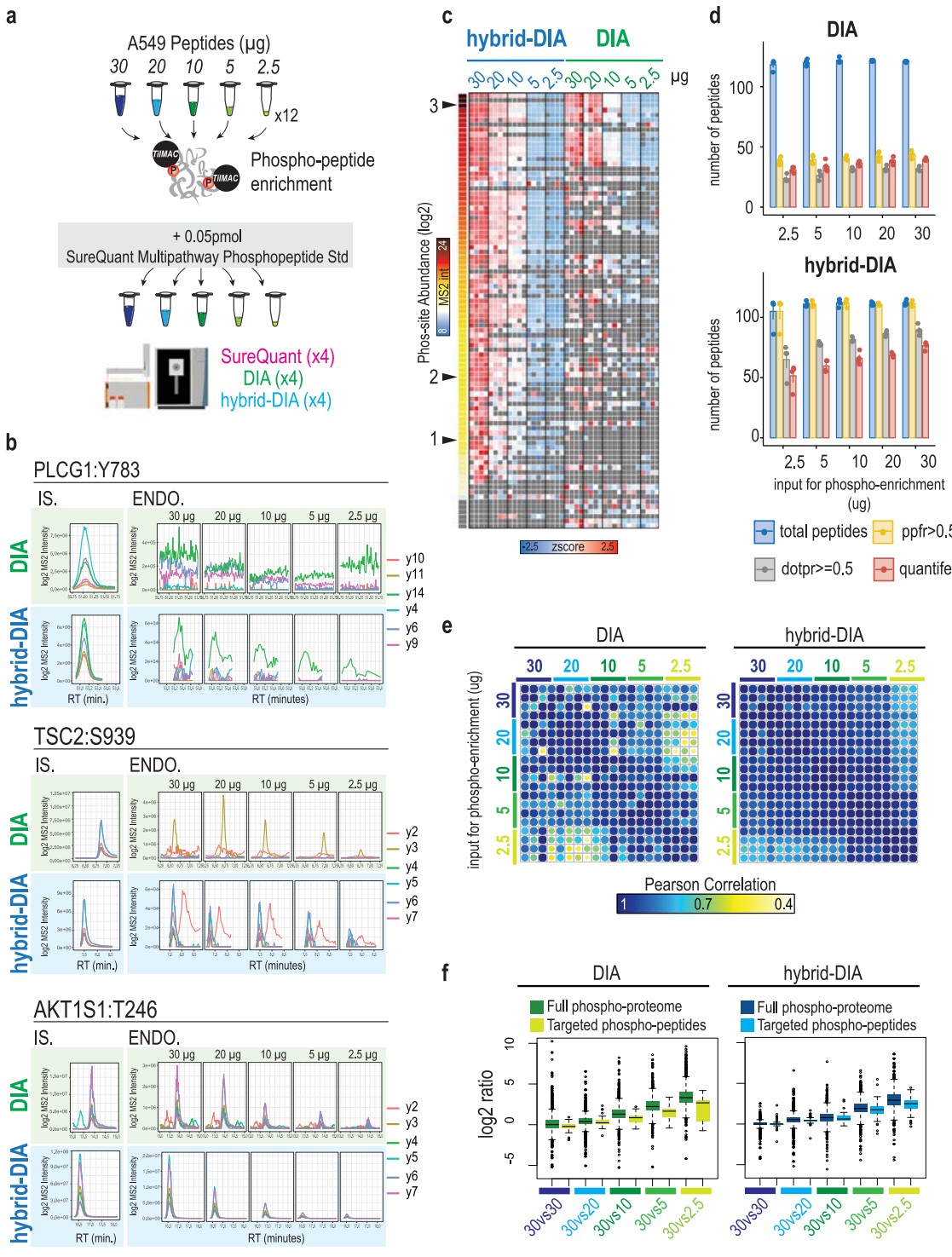

DIA and hybrid-DIA datasets (Fig. 4g) and found positive correlation that decreased with lower input amounts, which is comparable to the decreased found among standard-DIA replicates. This is likely due to the higher variability introduced when doing phospho-enrichment with very low phosphopeptide enrichment inputs (<10 µg) and due to the overall lower signal (Fig. 4g and Supplementary Figure 3B).

**Benchmark of hybrid-DIA against SureQuant for targeted analysis of EGF stimulation**

Having demonstrated the advantages of using hybrid-DIA compared to standard DIA runs, we next benchmarked its quantitative performance against the state-of-the-art spike-in triggered PRM acquisition

method named SureQuant[12]. A phosphopeptide-enriched A549 digest dilution experiment (Fig. 3a) was acquired using both SureQuant and hybrid-DIA methods in a way that we could directly compare the sensitivity of the targeted scans in both approaches. We observed that both methodologies provide equivalent quantification performance through the dilution series range in terms of precision and accuracy (Fig. 5a, b). As expected, the quantitative performance for both SureQuant and hybrid-DIA was affected by the relative abundance of the phosphopeptides, but still showing better sensitivity for low abundance phospho-sites than standard DIA (Fig. 3c). Standard DIA and SureQuant have their advantages when compared to hybrid-DIA, which has certain limitations (i.e., numbers of targets,

**Fig. 3 | Benchmark of hybrid-DIA versus standard DIA for sensitive phospho-proteomics analysis. a** Decreasing input material for phosphopeptides from A549 were used for phospho-enrichment ($n = 4$ replicates, 5 input amounts). Heavy-labeled peptide mixture was added, and samples were analyzed either by hybrid-DIA, DIA or SureQuant. **b** XIC for three phosphopeptides spanning the dynamic range of the phosphopeptide mixture. DIA runs in green, and data from MSx scans (corrected by injection time) in hybrid-DIA runs in blue. The three peptides represented in here correspond to peptides of different abundance in the sample as it is indicated as black arrows in **c**. RT: retention time, IS: heavy-labeled internal standard, ENDO: endogenous peptide. **c** Heatmap showing relative quantification (z-score intensities across samples) of targeted peptides in a dilution series experiment in hybrid-DIA (blue) and DIA (green). Sites are sorted by relative abundance. Black arrows indicate the position of the peptides used in **b** (3: AKT1S1-T246; 2: TSC2:S939; 1: PLGC1-Y783). **d** Barplots showing the number of targeted peptides detected and quantified in DIA (top) and hybrid-DIA (bottom) ($n = 4$ for each input amount, height of the bar indicates the mean of the four replicates, and the error bars the standard error). In blue, total number of heavy peptides detected when importing the data in Skyline. In yellow, number of heavy-endogenous pairs detected with a peptide-peak-found ratio or PPFR higher than 0.5. In gray, number of heavy-endogenous pairs with dot-product-ratio or DOTPR bigger or equal to 0.5. In red, number of heavy-endogenous pairs that can be quantified with at least 3 fragments. **e** Correlation plot of quantified heavy-endogenous ratios between replicates and input amounts for DIA (left) and hybrid-DIA (right) analysis. Correlation is measured by Pearson. **f** Boxplots showing the phospho-site ratio distribution between different conditions in DIA (left) and hybrid-DIA (right) analysis. Each plot shows the ratios between replicates of 30 μg of input material, 30 μg vs 20 μg, 30 μg vs 10 μg, 30 μg vs 5 μg and 30 μg vs 2.5 μg. Dark boxplots show the global distribution of all phospho-sites measured using the DIA windows in both methods and analyzed in Spectronaut (i.e: global phosphoproteome). Light box-plots show the distribution of the endogenous/heavy ratios. Center lines show the medians; box limits indicate the 25th and 75th percentiles as determined by R software; whiskers extend 1.5 times the interquartile range from the 25th and 75th percentiles, outliers are represented by dots. $N = 2666, 47, 2501, 50, 2153, 46, 1813, 40, 1390, 39, 2862, 84, 2450, 77, 2152, 76, 1599, 70, 1160, 63$. Source data are provided as a Source Data file.

MS method set-up and downstream analysis). However, overall, our benchmark highlights that hybrid-DIA offers the best compromise for performing both discovery and targeted analysis when compared to either DIA with heavy spiked-in peptides or SureQuant targeted analysis (Fig. 5c).

To extend this benchmark to a biologically interesting cell signaling scenario, we performed EGF stimulation and chemical inhibition of downstream kinases in HeLa cells as a model of dynamic cellular signaling pathway rewiring (Fig. 6a). We selected this model system because the SureQuant Multipathway Phosphorylation Mix panel covers the EGFR signaling pathway as well as the main downstream MEK and PI3K kinase pathways. Moreover, to test both methods in the most challenging conditions with limited input material, we also scaled down the input material growing cells in P6 plates to obtain approximately 50 μg of peptide per condition prior to phosphopeptide-enrichment.

The goal here was to use the targeted data to reconstruct the phosphorylation pathways and infer the inhibited kinases (Fig. 6b). When comparing the quantitative profiles of the targeted peptides in SureQuant and hybrid-DIA of the sites differentially regulated by 10 minutes EGF stimulation, we observed that both methodologies provide equivalent results (Fig. 6b, c, Supplementary Fig. 4 and Supplementary Data 1A, B). Interestingly, with both SureQuant and hybrid-DIA and using the panel of synthetic heavy phosphopeptides, we can clearly identify kinase-specific responses with EGFR and AKT sites dynamically regulated by both EGF and kinase inhibitors, reflecting the potential of both methodologies to recapitulate kinase activity using this panel of peptides (Fig. 6b, c). Furthermore, the main advantage of hybrid-DIA over SureQuant, is that hybrid-DIA data also provided quantitative data related to the background phosphoproteome covering 6,291 sites (Fig. 6d), on top of the quantification of the sites from the panel. Using this data in a discovery phosphoproteomics pipeline, we could further reinforce the information on kinase inference (Fig. 6e, Supplementary Data 2). Using RoKAI[21], a computational tool for inferring kinase activities, we identified a rapid activation of multiple kinases upon EGF stimulation and how this activity was abrogated when inhibiting EGFR with lapatinib (Fig. 6e). Furthermore, we observed that MTOR activity, which is a downstream target of PI3K, is specifically inhibited with lapatinib (EGFRi) and wortmannin (PI3Ki), but not by PD0325901 (MEKi). Conversely, MAPK1 signaling, the direct target of MEK, is significantly reduced after PD0325901 (MEKi) treatment but not affected by wortmannin (PI3Ki) (Fig. 6e). Collectively, these results show the advantages of performing hybrid-DIA rather than only targeted acquisition methods as it maximizes the information retrieved from single-shot phosphoproteomics samples.

## Phosphoproteomics signature in 2D vs 3D model of colorectal cancer

Finally, we decided to apply our intelligent data acquisition strategy to the most challenging biological in vitro models by studying dynamic phosphoproteome signaling in single multicellular cancer spheroids, and compare the signaling in these to conventional 2D-monolayer culture of colorectal cancer cells. Three-dimensional tumor models, such as spheroids, offers an improved model to assess molecular and physiological aspects that are essential for drug development including drug penetration, hypoxic/necrotic environment, stemness and cell interaction, among many others[22,23]. Traditionally, spheroid models have been technically challenging, especially from a proteomics perspective due to the low protein amount obtained from single spheroids, which typically requires pooling of several spheroids per condition to achieve a reasonable proteome coverage[24–26]. These limitations are even more evident when studying the phosphoproteome layer, due to the need for phosphopeptide-enrichment prior to LC-MS/MS measurements. Consequently, we reasoned that drug screening in single spheroids by phosphoproteomics was an ideal example of an experimental set-up requiring high sensitivity and benefitting from using our hybrid-DIA pipeline. To achieve as good phosphopeptide coverage in single spheroids as possible, we have improved the sensitivity of our phosphoproteomics pipeline with the introduction of a modified phospho-enrichment protocol in combination with higher-resolution online chromatography to enhance MS sensitivity. For the latter, we took advantage of the higher sensitivity and chromatographic performance achieved with Whisper nanoflow gradients on the Evosep One LC platform when using the Aurora column from IonOpticks (Supplementary Fig. 5A). Next, we observed that the use of MagReSyn® ZrIMAC-HP beads[27] outperformed MagReSyn® Ti-IMAC-HP beads for low peptide input amounts in the low microgram range (Supplementary Fig. 5B). Additionally, we previously described how a second phosphopeptide-enrichment step in the Kingfisher Flex automated platform is easily implemented by looping through the protocol, without the need to change buffers, but also reusing the beads and the elution buffer[28]. The implementation of the improved experimental protocol in combination with hybrid-DIA MS analysis, maximized the phospho-signaling information retrievable from single spheroids.

We decided to apply this improved pipeline to investigate phospho-signaling in single spheroids and compare their response to cells grown in monolayer culture in the context of sensitivity to a chemotherapeutic agent active against colorectal cancer. For that purpose, we used either a 2D model (monolayer adherent cells) or a 3D model (single multicellular spheroids) that we treated with 5-fluorouracil (5-FU) (Fig. 7a). We employed the HCT116 colorectal cell

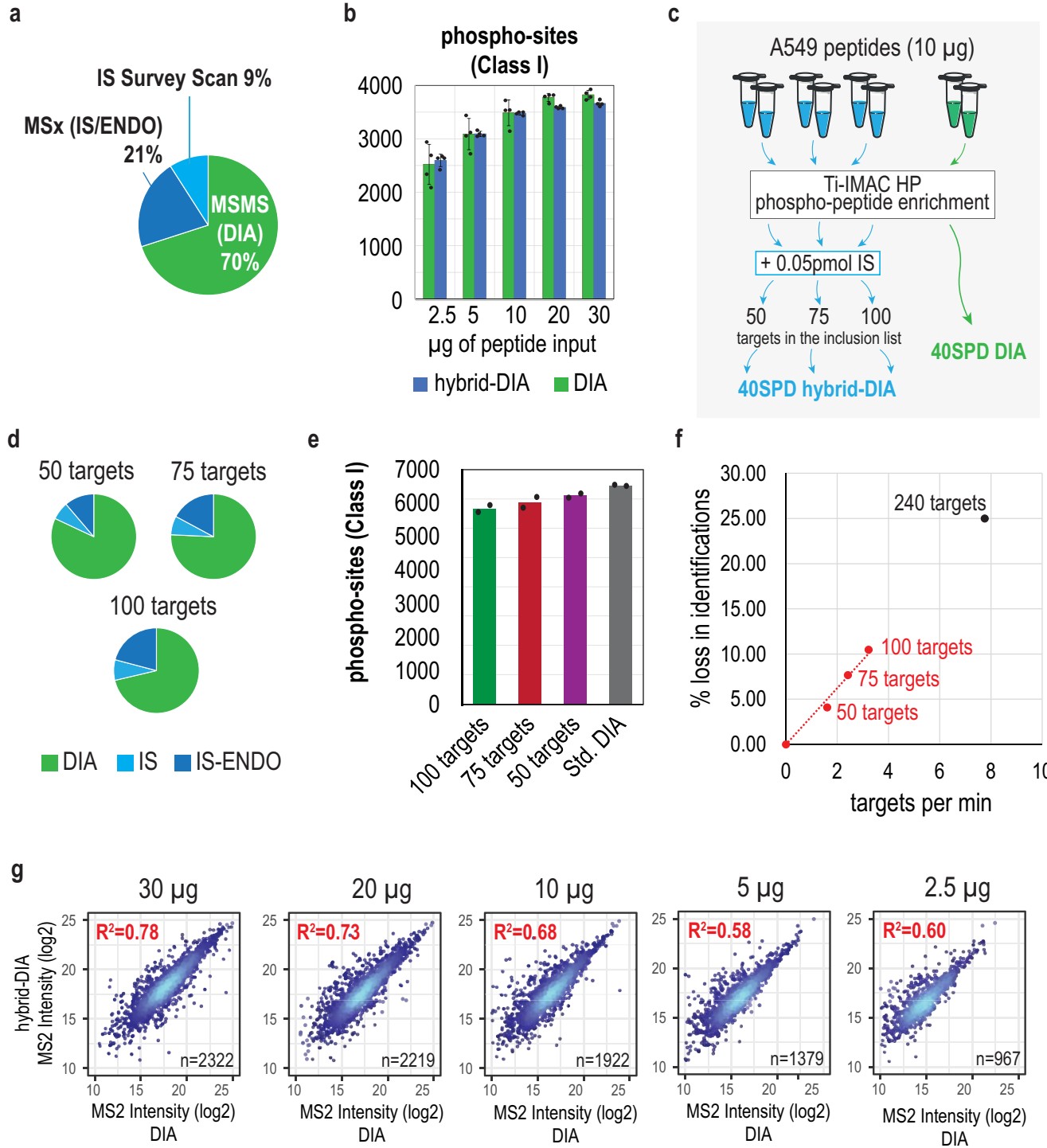

**Fig. 4 | Evaluation of the depth achieved from DIA data from hybrid-DIA datasets. a** Pie charts representing the proportion of cycle time used by the hybrid-DIA API measured in acquisition time. **b** Number of phospho-sites (class I) identified using Spectronaut (v17) in DIA (green, *n* = 4 independent experiments) and hybrid-DIA (blue, *n* = 4 independent experiments). Length of the bars indicate the average of the experimental replicates, and the error bars the standard deviation. **c** Experimental design to evaluate the effect of different number of peptides used in the inclusion list during hybrid-DIA analysis. **d** Pie charts representing the proportion of cycle time used by the hybrid-DIA API measured by acquisition time when using an inclusion list of 50, 75 or 100 targets. In green, the proportion of the total MS2 acquisition time used in DIA scans; in light blue, the time used in survey scans for detecting the presence of the internal standard (IS) and in dark blue, the time used in multiplexed (IS and ENDO peptides). **e** Barplots

showing the number of phospho-sites (class I) identified in either standard DIA runs, or when using hybrid-DIA methods and increasing number of targets. Bar length represents the average of the identified sites between replicates (*n* = 2). **f** Relationship between peptides targeted per minute (calculated by dividing the total number of peptides in the inclusion list by the gradient length) against the percentage of lost identifications (using standard DIA values as reference). In black, hypothetical extrapolation of the number of targets that would lead to 25% of lost identifications. **g** Correlation plot of quantified phospho-sites in hybrid-DIA runs (y-axis) versus DIA runs (x-axis) for the different dilutions. Correlation is indicated as R-squared. IS: heavy-labeled internal standard, IS/ENDO: targeted multiplex scan with internal standard and endogenous peptide. Source data are provided as a Source Data file.

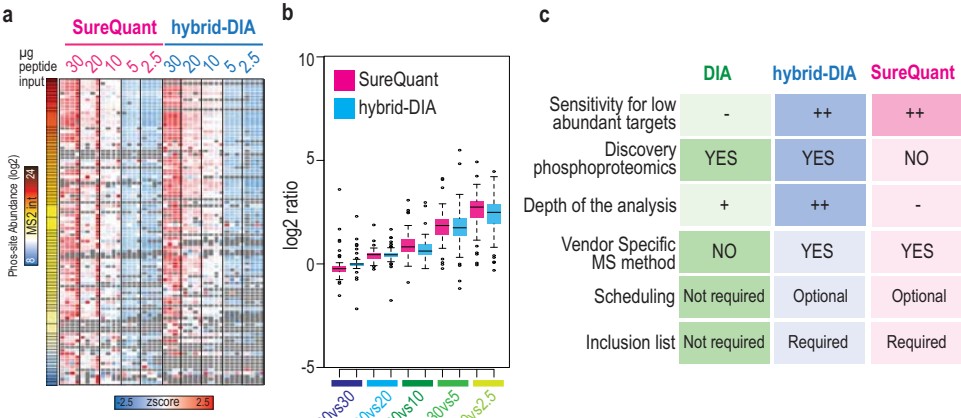

**Fig. 5 | Benchmark of hybrid-DIA versus SureQuant for sensitive phospho-proteomics analysis. a** Heatmap showing relative quantification of targeted peptides in a dilution series experiment (Fig. 3a) in SureQuant (pink) and hybrid-DIA (blue). Sites are sorted by abundance. **b** Boxplots showing the phospho-site ratio of targeted peptides (measured as log2 of the ratio ENDO/IS in one amount versus ratio ENDO/IS in 30 µg) in SureQuant (left) and hybrid-DIA (right) analysis. Each plot shows the ratios between replicates of 30 µg of input material, and 30 µg vs 20 µg, 30 µg vs 10 µg, 30 µg vs 5 µg and 30 µg vs 2.5 µg. Center lines show the medians; box limits indicate the 25th and 75th percentiles as determined by R software; whiskers extend 1.5 times the interquartile range from the 25th and 75th percentiles, outliers are represented by dots. N = 80, 84, 82, 77, 79, 76, 76, 70, 72, 63. **c** Comparative table of characteristics and performance of DIA, hybrid-DIA, and SureQuant based on different aspects of shot-gun and targeted proteomics analysis. Source data are provided as a Source Data file.

line (ATCC: CCL-247), which has previously been employed to study differential responses[29] between 2D and 3D grown cells, and HCT116 has shown sensitivity to 5-FU[30], which is the first-line treatment for adjuvant therapy for colorectal cancer in the clinic[31,32]. For both monolayer and 3D culture, we seeded 20,000 cells per condition and grew them for three days until the spheroids were fully formed. At that time, 5-FU was added and samples collected at 0, 1, 3, 6, 12, and 24 hours after treatment with each condition as five independent replicates. Overall, the experiment consisted of 30 single spheroids and 30 samples of monolayer counterparts (Fig. 7a). All samples were lysed in 5% SDS, proteins extracted and trypsin digested using the protein aggregation capture (PAC) protocol, and phosphopeptides enriched using ZrIMAC HP beads. To each of the resulting 60 phosphopeptide samples, we spiked-in 50 fmol of the SureQuant Multi-Pathway Phosphorylation kit, which contained several cellular phosphorylation site markers of DNA damage and apoptosis, such as HSPB1:S82[33], HSPB1:S15, JUN:S63[34] and TP53:S315[35,36]. From the resulting raw MS files, we extracted the data from all targeted IS/ENDO MSx scans, and found 62 phosphorylation sites that were differentially regulated in at least one time point in either of the two conditions (3D-spheroids or 2D-monolayer) (Fig. 7b, Supplementary Data 3A–D). Although the data from the targeted analysis revealed a similar response in terms of 5-FU-activated signaling pathways in 2D-monolayer cells and the 3D spheroid model, there were some notable differences. Substrate sites of the stress-responsive kinase, MAPKAPK-2 (MK2), HSPB1 Ser15 and Ser82 were phosphorylated at 6 to 12 hours in monolayer culture, whilst their upregulation required up to 24 hours in spheroids (Fig. 7b, c). In contrast, apoptosis-activating phosphorylation sites on JUN Ser63 and TP53 Ser315 showed more synchronous temporal profiles in both systems, with significant activation as early as 3 hours for the Jun phosphorylation site (Fig. 7b, c). Interestingly, we also found phospho-sites related to MTOR and GSK3 signaling, which were specifically upregulated in spheroids only peaking at the latest 24 hours time point (Fig. 7b). The targeted analysis of this panel of phosphopeptides, therefore, serves as a highly sensitive and multiplexed assay to accurately probe the activity state and signaling dynamics of the major cellular kinase pathways directly informing about the signaling state of the cells analyzed.

Furthermore, in addition to the phospho-signature extracted by the targeted quantification of the phosphopeptide panel, the hybrid-

DIA MS data also contained comprehensive phosphoproteome profiling from the DIA scans. After conversion of the files to HTRMS format, we analyzed them with directDIA in Spectronaut (v17) obtaining quantification for 18,946 localized phospho-sites. To perform quantitative comparisons, we filtered the global phosphoproteomics dataset and retained 8,783 phospho-sites that were quantified in at least three out of five spheroids analyzed per one treatment time point, whilst 12,084 phospho-sites were quantified in the same proportion of samples in the monolayer culture condition (Fig. 8a, Supplementary Data 4A, B). Such a phosphoproteome coverage is on par with data obtained in other large-scale phospho-proteomics screenings that use significantly higher peptide input material[5,28,37]. This reflects that the improvements in our sample preparation processing and MS analysis pipeline makes it realistic to scale down input amounts for highly sensitive phosphoproteomics experiments, such as analyzing single spheroid, while preserving a considerable coverage of the quantifiable phosphoproteome. Phosphoproteomics profiling of each sample reflected the expected biological differences between 2D and 3D grown cells, such as an increase in cell-cycle activation in 2D cells[38] or lowered RPS6 phosphorylation in spheroids[39] (Supplementary Fig. 6A, B).

Principal Component Analysis (PCA) showed a clear separation between monolayer cells and spheroids in the first principal component (PC1), whereas the temporal effects of 5-FU followed similar trends in both conditions in principal components two and three (PC2 and PC3) (Fig. 8b). The 5-FU mechanism of action impairs DNA replication by inducing double-strand breaks (DBSs) during S phase of the cell-cycle activating the DNA damage response[40–42]. Accordingly, we observed that phosphorylation of serine 140 in histone H2AX, a biomarker of DSBs known as gamma-H2AX, increases significantly upon treatment of colorectal cells with 5-FU (Fig. 8c). Interestingly, baseline levels of H2AX Ser140 were slightly higher in spheroids compared to adherent 2D-monolayer-cultured cells, but, in contrast, monolayer-grown cells showed significantly higher increase in the phosphorylation site change of this marker when compared to spheroids, especially evident at 12 hours of treatment (Fig. 8c). This could indicate that monolayer-grown cells are more sensitive to the effect of 5-FU than spheroids, and highlights the importance of using 3D models exhibiting a different sensitivity to chemotherapeutic agents than cells grown in plates. The observed discrepancy in drug response kinetics is

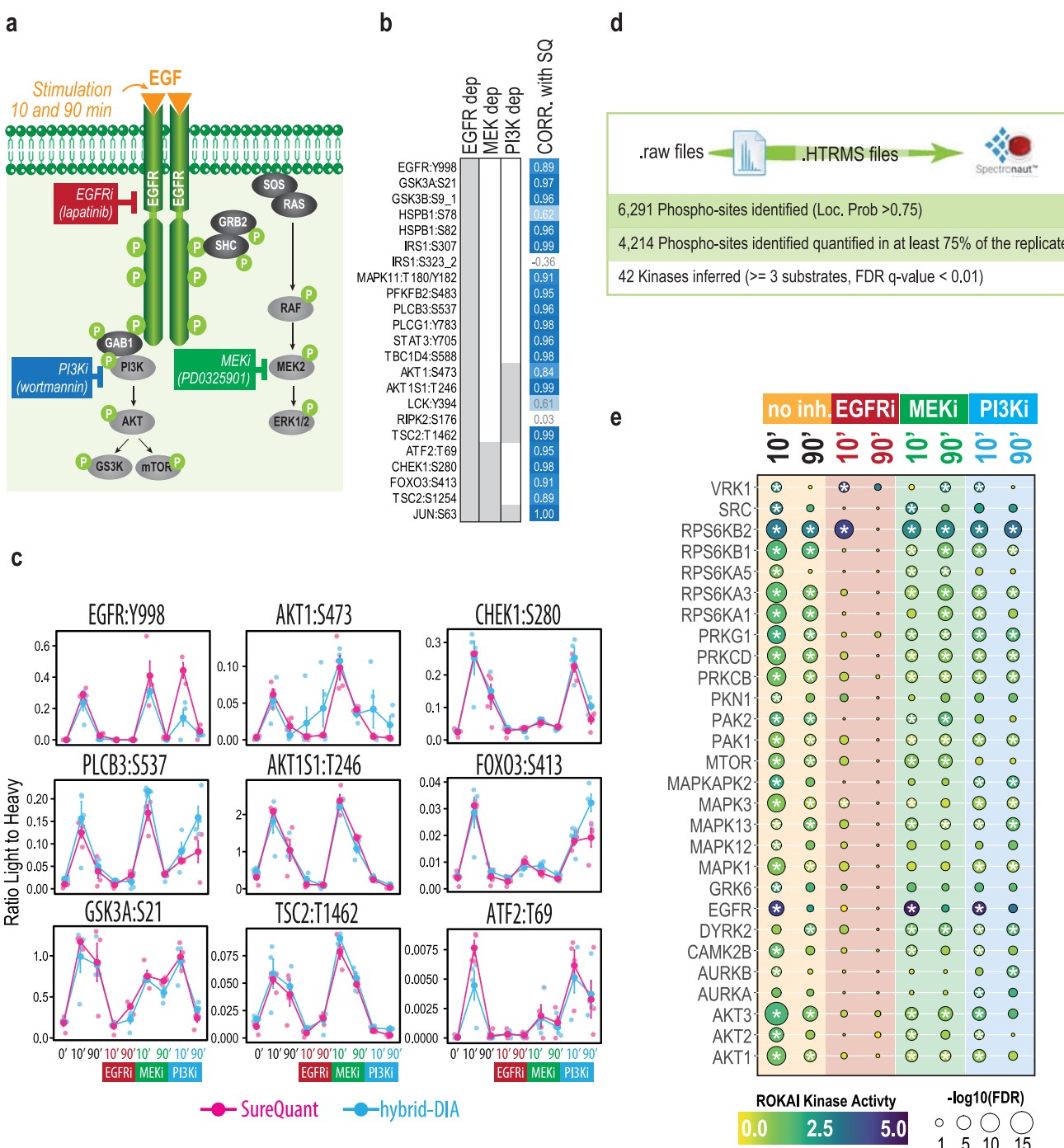

**Fig. 6 | Comparison of hybrid-DIA and SureQuant for EGF signaling pathway reconstruction. a** Experimental design to study EGF time course stimulation in the presence of three inhibitors of downstream pathways: EGFRi (lapatinib), PI3Ki (wortmannin) and MEKi (PD0325901). Each condition was performed in quadruplicates. **b** Significant proteins after 10 min of EGF stimulation (two-sided, two-sample *t*-test, log2 FC (10 min vs control) >1.5, *q* value < 0.01, *n* = 4 biological replicates) that are not upregulated after EGFR inhibition. MEK dependent are those regulated after 10 min of EGF stimulation that do not show upregulation after PD0325901 treatment. PI3K dependent are those regulated after 10 min of EGF stimulation that do not show upregulation after wortmannin treatment. Gray indicates that the site was found to be dependent on the indicated kinase. Last column show the Pearson correlation between the profile of those sites in hybrid-DIA and SureQuant. **c** Profile plot of representative regulated sites by EGF (two-

sided, two-sample t-test, log2 FC (10 min vs control) >1.5, q-value < 0.01). Dots indicate the absolute ratio Endogenous to Heavy standard. In blue, data from hybrid-DIA quantification (*n* = 4 biological replicates); and in pink, data from SureQuant quantification (*n* = 4 biological replicates). Line indicate the average of the experimental replicates, and the error bars the standard error of the mean. **d** Summary of results obtained from the extracted DIA scans from the hybrid-DIA experiment. **e** Kinase activity inference analysis obtained from RoKAI using the discovery analysis data from the hybrid-DIA experiment. Color of the dots reflect the value of the inferred kinase activity calculated by ROKAI21 (higher values correlate with higher kinase activity in the sample). Size of the dots is proportional to the -log10 of the FDR corrected p-value. Asterisks indicate FDR *q* value < 0.01. Source data are provided as a Source Data file.

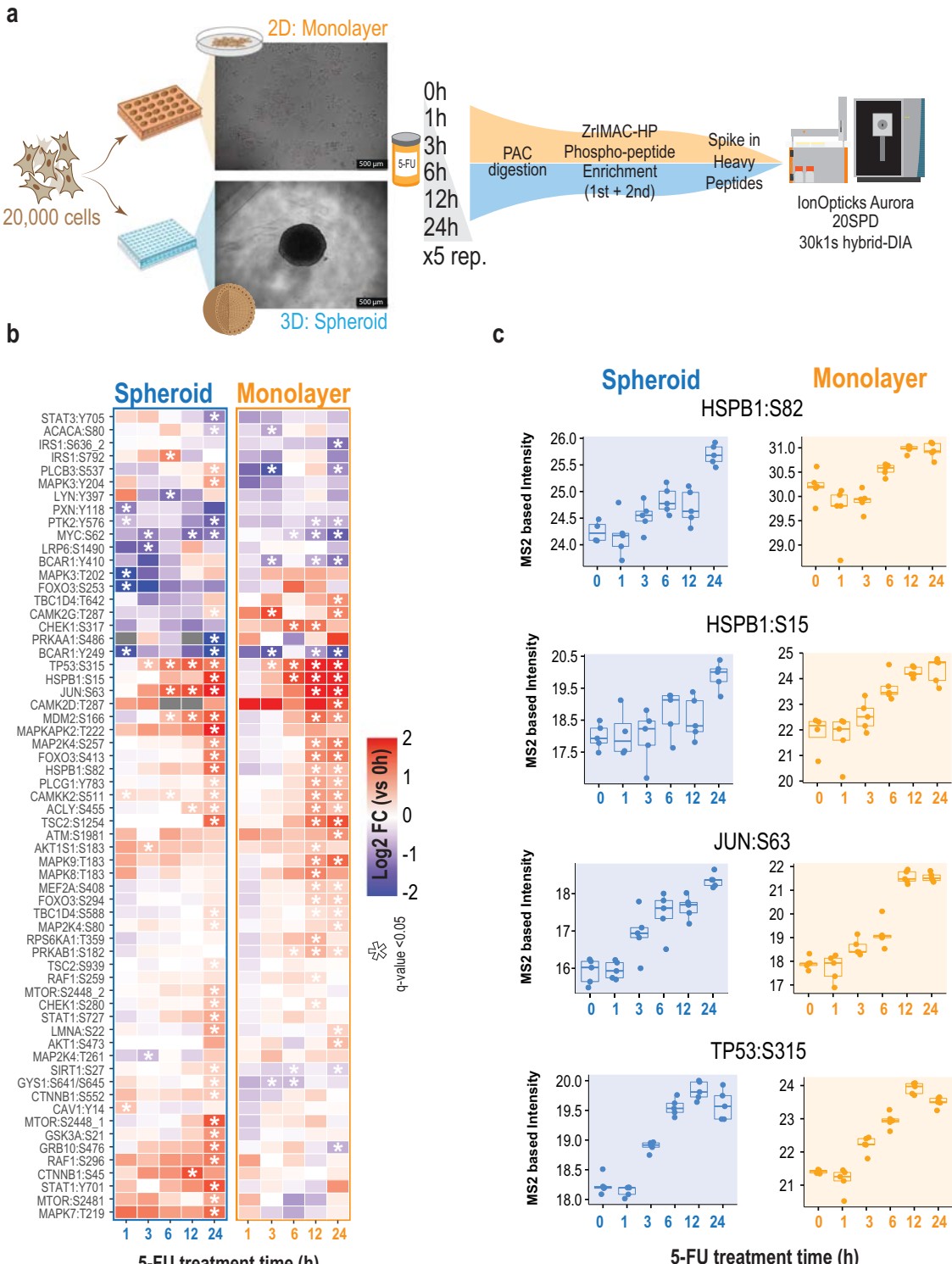

**Fig. 7 | Hybrid-DIA targeted analysis of response to 5-fluouracil in cell culture models for colorectal cancer. a** Experimental design for the comparison of spheroids against monolayer culture of HCT116 cancer cells treated with 5-Fluorouracil. **b** Heatmap showing the phosphosites from the SureQuant™ Multipathway Phosphopeptide Standard panel that are differentially regulated (two-sided two-samples t-test, BH-FDR) in at least one point. Color indicates the average log2 fold change of each time point against time 0 (*n* = 5 biological replicates). Asterisk indicates q-value < 0.05. **c** Boxplot of MS2 intensities obtained from

hybrid-DIA scans of relevant phosphorylation markers of DNA damage (*n* = 5 biological replicates) in single spheroids or monolayer grown cells. Intensities are not normalized by loading amount between spheroids and monolayer cells. Center lines show the medians; box limits indicate the 25th and 75th percentiles as determined by R software; whiskers extend 1.5 times the interquartile range from the 25th and 75th percentiles, outliers are represented by dots. Source data are provided as a Source Data file.

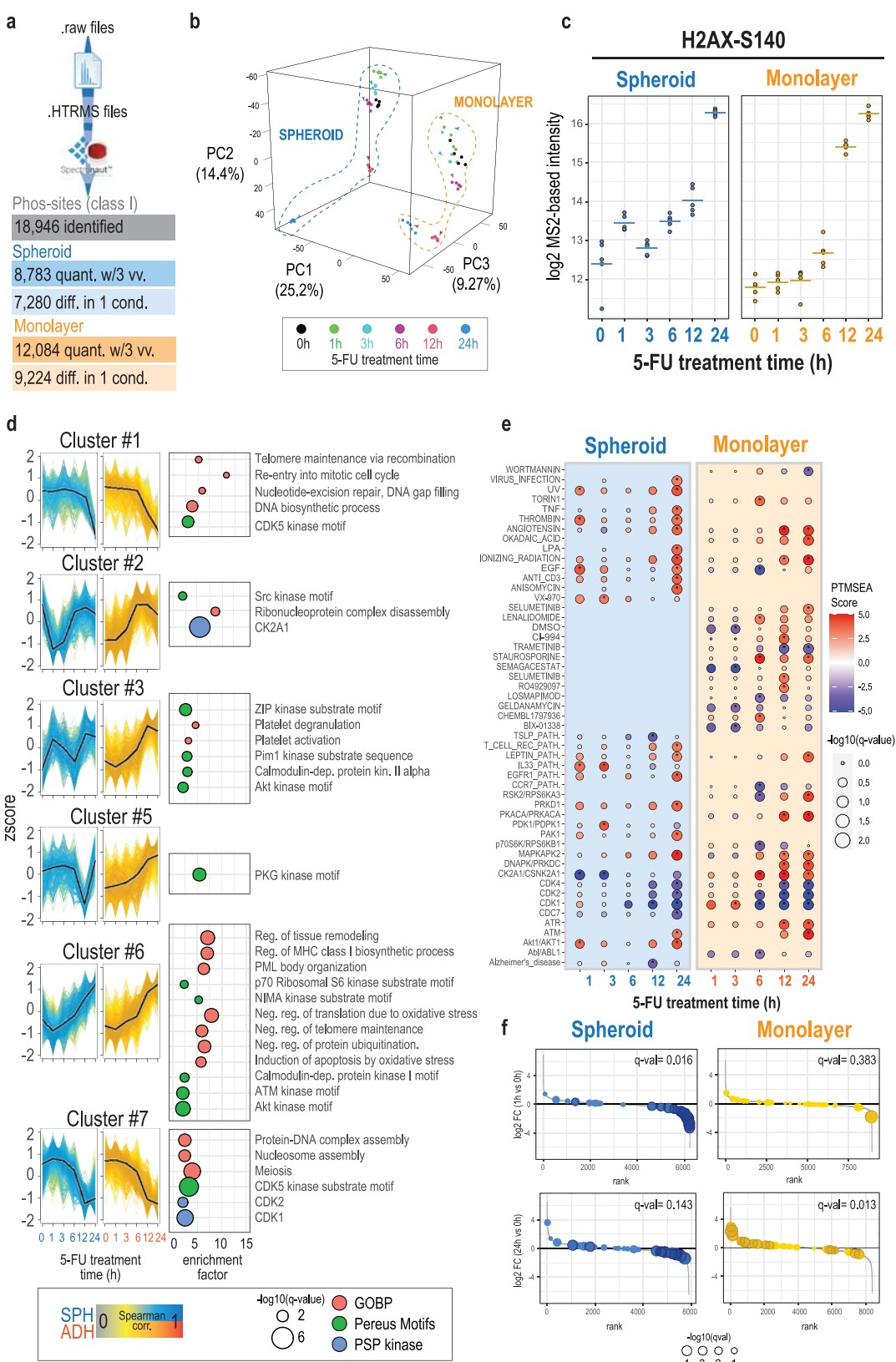

probably due to inherent delay in the diffusion of the drug into the inner spheroid in order to exert its full effect[43]. Furthermore, to identify 5-FU-activated kinase signaling pathways in an unbiased manner, we performed an exploratory bioinformatics analysis of the full DIA phosphoproteomics dataset. We clustered the regulated phospho-peptide sites after z-scoring across the different drug-treatment time

points and extracted the different temporal profiles observed in both treatments (Supplementary Fig. 6C, D). This analysis revealed that the main biological response triggered by 5-FU are equivalent in both cell models: clusters #1 and #7 shows the downregulation of cell-cycle control by CDKs, and, conversely, cluster #6 shows a parallel upregulation of signaling pathways related to stress mediated by ATM and

**Fig. 8 | Discovery phosphoproteomics analysis using hybrid-DIA allows to describe temporal response and kinase activation after 5-fluorouracil treatment in 2D and 3D model. a** Overview of the results obtained from the analysis of the DIA data with Spectronaut, after conversion to HTRMS format. **b** Principal Component analysis of spheroids and monolayer-grown cells treated with 5-FU at 0, 1, 3, 6, 12, and 24 h (n = biological replicates 5). **c** Log2 intensity at MS2 level of H2AX Serine 140 (n = biological replicates 5, horizontal lines indicate the average of all measures). **d** Temporal profiles for relevant clusters (see Supplementary Figure 6C). To the right of each graph, the results from a Fisher's exact test to show overrepresentation of terms from GOBP, Phosphositeplus Kinases and Kinase motifs. Size of the dot indicates the significance (two-sided, Fisher's exact test, BH-

FDR corrected), position on the x-axis, the enrichment factor, and the color indicates the ontology to which each term belongs. **e** PTMSEA results. Size of the dot indicates the significance (BH-FDR corrected); color indicates whether the term (associated pathway or kinase) is upregulated (red) or downregulated (blue). **f** Rank plots showing phospho-sites ranked by their fold change (log2) at 1 or 24 h of treatment versus non-treated samples. Dots indicate the position of phospho-sites from the CK2A1 term (from PTMSEA database). Size of the dot indicates the significance of the fold change (limma robust moderated t-test, two-sided, BH-FDR, n = biological replicates 5). Darker dots highlight the sites with FDR corrected p-value < 0.01. Source data are provided as a Source Data file.

AKT kinases (Fig. 8d). However, this analysis also showed significant differences between the two cell models, for instance, the distinct temporal trend in the upregulation of AKT, or the specific early downregulation of CK2A1 in spheroids. We complemented this bioinformatics analysis using PTMSEA[44] to annotate different phospho-regulated pathways and kinases in either 3D or 2D models across 5-FU treatment time. As expected, we found that MAPKAPK2 kinase, a stress-responsive kinase that has been connected with resistance in 5-FU-treated colorectal cancer cells[45], was upregulated by treatment with 5-FU. In line with the targeted phosphopeptide data, the activation of this kinase was evident earlier (12 h) in the monolayer cells than in the spheroids (Fig. 8e). On the other hand, we found that cyclin-dependent kinases 1, 2, and 4 activities were strongly downregulated after 5-FU treatment in both conditions but more significantly in monolayer cells (Fig. 8e). Interestingly, upregulation of CDK1 and CDK4 is correlated with poor prognosis in cancer patients, especially in those with resistance to 5-FU[46–48]. Although most of the phospho-site signatures follow the same trend in spheroids and monolayer culture, we found casein kinase 2 alpha (CK2A1) to be strikingly different with opposite regulation (Fig. 8d, e). There is evidence that CK2A1 levels correlate with poor prognosis in CRC patients[49]. Here we found a strong and rapid downregulation of this kinase in the spheroids, whilst it shows a slight upregulation in monolayer-grown cells (Fig. 8f).

## Discussion

In this work, we present a MS acquisition method, termed hybrid-DIA, which enables intelligent acquisition of a predefined set of target peptides while acquiring shotgun proteomics data using DIA in the same cycle (Supplementary Note 1). Consequently, our methodology combines accurate and sensitive quantification of targeted proteomics with the depth and unbiased discovery analysis of traditional DIA methods (Fig. 5c). Importantly, to facilitate the usage and analysis of data derived from using this method, we have designed a freely available data analysis pipeline (Supplementary Note 2).

We assume that this acquisition method is most beneficial for biomedical applications where sample input is limited and/or high-throughput is requested for large sample size analysis, and therefore, it is critical to maximize the information that can be retrieved from a single-shot MS run. To serve as examples of such applications, we have performed highly sensitive phosphoproteomics of model systems with limited material using hybrid-DIA to prove the benefits of this workflow. As demonstrated by our data, hybrid-DIA benefits are more significant when targets of interest are of low abundance. This is because the targeted part of the method improves the limit of detection and quantification of predefined targets, which in standard DIA analysis will not be confidently detected.

One important consideration for the implementation of the hybrid-DIA workflow is that the inclusion of targeted scans in the DIA acquisition scheme is accompanied by prolonged MS acquisition cycle time, which needs to be carefully assessed when defining the number of targets. Ideally, the peptide targets are evenly distributed across the chromatographic elution time range such that the number of targets per minute is regular through the run. Moreover, we presented

guidelines on how to design the experiment based on the number of targets and the expected reduction in the depth of the proteome. For instance, in our set-up we estimated that 8 targets per minute would potentially lead to a reduction of 25% in overall DIA-based identifications (Fig. 4f). Therefore, to prevent such losses it would be important to elongate the chromatographic gradient when scaling up the target list.

Most interestingly, not only have we demonstrated that both the DIA and targeted parts of the hybrid-DIA workflow are on par with state-of-art proteomics for each method, but also the usefulness of hybrid-DIA methodology for clinical research purposes. Clinically relevant in vitro models in early drug screening are essential for developing potent and effective chemotherapeutic agents. Three-dimensional tumor models such as cancer cell spheroids more closely mimic in vivo solid tumors than monolayer-cultured cell lines. These 3D cell models have emerged as attractive models for the early stages in drug screening due to their similarity to in vivo tumor tissue, metabolically and regarding proliferation and gradient distribution[50,51]. However, due to the nature of spheroids, they are limited in size, and therefore in the amount of protein available for subsequent MS analysis[52]. Previous proteomics investigations relied on pooling of multiple single-spheroids for each condition analyzed[53], which significantly reduces the throughput of this model. This limitation is even more relevant when studying the phosphoproteome, and to our knowledge, there is no prior work that provides comprehensive phosphoproteome profiles of single spheroids. However, it is also important to highlight that, when studying the effects of drugs on spheroids and comparing to cells grown in monolayer, the drug adsorption into the spheroid core and the effect of the drug into the parenchyma can increase the variability and heterogeneity when measuring full single-spheroids signaling by MS. These limitations could potentially be addressed by studying the different layers of the spheroid as separate fractions[53,54]. Nevertheless, without doubt, a methodology that allows assessing phosphoproteomic response in single spheroids will truly increase the throughput of drug screening, and aid the field in the direction of using 3D models instead of 2D-monolayer grown cellular models.

## Methods

### Statistics and reproducibility

No sample size calculation was performed. A number of replicates was chosen based on previous expertize to obtain enough statistical power. All proteomics experiments were performed in replicates. Four experimental replicates (workflow replicates) were used for method benchmarking. Four biological replicates (four cell dishes) were employed for EGF stimulation in HeLa cells. Five biological replicates (separate cell dishes or single spheroids) were used in the time course treatment with 5-fluorouracil in HCT116 cells. Allocation of samples, cells or spheroids to each experimental group was random. In the EGF stimulation experiment with HeLa cells, replicate number 2 from 10 min treatment of Inhibitor 1 was excluded from DIA analysis due to poor identification rate. The investigators were not blinded to allocation during experiments and outcome assessment.

## Sample Prep: A549 dilution series for phosphoproteomics

A549 (ATCC CCL-185) were cultured in DMEM (Gibco, Invitrogen), supplemented with 10% fetal bovine serum (FBS, Gibco), 100U/ml penicillin (Invitrogen), 100 µg/ml streptomycin (Invitrogen), at 37 °C, in a humidified incubator with 5% CO2. Cells were harvested at ~80% confluence by washing twice with PBS (Gibco, Life Technologies) and subsequently adding boiling lysis buffer (5% sodium dodecyl sulfate (SDS), 5 mM tris(2-carboxyethyl)phosphine (TCEP), 10 mM chloroacetamide (CAA), 100 mM Tris, pH 8.5) directly to the plate. The cell lysate was collected by scraping the plate and boiled for an additional 10 min followed by micro tip probe sonication (Vibra-Cell VCX130, Sonics, Newtown, CT) for 2 min with pulses of 1 second on and 1 second off at 80% amplitude. Protein concentration was estimated by BCA.

Protein was digested using the Protein Aggregation Capture[55] protocol in the KingFisher robot. Briefly 1 mg of protein was resuspended with acetonitrile to a final 70% concentration. MagReSyn® Hydroxyl beads were added in a proportion 1:2 (protein:beads). Protein aggregation was performed in two steps of 1 min mixing at 1000 rpm, followed by 10 min pause each. Beads were subsequently washed three times with 1 ml 95% ACN and two times with 1 ml 70% EtOH. 300 µl of digestion buffer (50 mM Ammonium Bicarbonate) and proteases were added in the following proportions: trypsin 1:250 (enzyme:protein) and lysC 1:500 (enzyme:protein). Digestion was carried out overnight at 37 °C with looping mixing. Protease activity was quenched by acidification with trifluoroacetic acid (TFA) to a final concentration of 1%, and the resulting peptide mixture was concentrated on Sep-Pak (C18 Classic Cartridge, Waters, Milford, MA). Peptides were eluted with 150 µl 40% ACN, followed by 150 µl 60% ACN. The combined eluate was reduced by SpeedVac (Eppendorf, Germany) and the final peptide concentration was estimated by measuring absorbance at 280 nm on a NanoDrop 2000C spectrophotometer (Thermo Fisher Scientific). For phosphoproteomic enrichment, each peptide amount (30, 20, 10, 5, and 2.5 µg) were resuspended with 200 µl of Loading buffer (80% ACN; 5%TFA, 1 M Glycolic Acid). Subsequent phospho-enrichment was performed in the King-fisher robot using 5 µl of MagReSyn® Ti-IMAC HP beads (20 mg/ml)[16]. Briefly, the 96-well comb is stored in plate #1, 10 µl Tii-IMAC HP beads in 100% ACN in plate #2 and loading buffer (1 M glycolic acid, 80% ACN, 5% TFA) in plate #3. Plates 5–7 are filled with 500 µl washing solutions; WB1 (Loading buffer), WB2 (80% ACN, 5% TFA), and WB3 (10% ACN, 0.2% TFA) respectively. Plate #8 contains 200 µl 1% ammonia for elution. The beads were washed in loading buffer for 5 min at medium mixing speed, followed by binding of the phosphopeptides for 20 min and medium speed. The sequential washes were performed in 2 min and fast speed. Phosphopeptides were eluted in 10 min at medium mixing speed. Enriched phosphopeptides were acidified with 10% TFA until pH < 3 and filtered to remove in-suspension particles (1 min, 500 × $g$, MultiScreenHTS HV Filter Plate, 0.45 µm, clear, non-sterile). 0.5 µl of the SureQuant™ Multipathway Phosphopeptide Standard (100 fmol/µL) was added to each sample prior loading into Evotips for subsequent MS analysis.

## Sample Prep: HeLa stimulation with EGF and kinase inhibitors

HeLa (ATCC CCL-2) cells were grown in a P6 dish until 70% confluence. Cells were serum-starved for 6 h. Control HeLa cells were stimulated with 100 ng/mL of EGF for 10 and 90 min. For the drug inhibitor treatment, cells were initially incubated in each inhibitor (Lapatinib 14 µM, PD0325901 3 µM, and Wormannin 25 µM) for 15 min. Then, cell were stimulated with 100 ng/mL of EGF for 10 and 90 min, in the presence of the inhibitors.

Cells were lysed with 200 µl of boiling lysis buffer (5% SDS; 100 mM Tris pH 8.5, 5 mM TCEP, and 10 mM CAA) and incubated at 95 °C, for 10 min with mixing (1000 rpm). Lysates were sonicated with an 8-tip probe (1 min, 1 second on, 1 second off, 50% amplitude, 8-channel Fisherbrand™ Tip Horn for Model 120 Sonic Dismembrator). Protein concentration was calculated by BCA. 150 µg of protein was digested using the Protein Aggregation Capture protocol in the King-fisher Robot as detailed above. Digested peptides were acidified after digestion with TFA to a final 1% concentration and loaded into a Sep-Pak tC18 96-well Plate, (40 mg Sorbent per Well, Waters) for desalting. Peptides were eluted in 75ul of 80% ACN and resuspended with 150 µl of Concentrated Loading buffer (80% ACN; 8%TFA, 1.6 M Glycolic Acid). 0.5 µl of the SureQuant™ Multipathway Phosphopeptide Standard (100 fmol/µL) was added to each sample and continued for subsequent phospho-enrichment in the Kingfisher robot using 5 µl of MagReSyn® Ti-IMAC HP beads (20 mg/ml). Enriched phosphopeptides were acidified with 10% TFA until pH < 3 and filtered to remove in-suspension particles (1 min, 500 × $g$, MultiScreenHTS HV Filter Plate, 0.45 µm, clear, non-sterile). Finally, samples were loaded into Evotips for subsequent MS analysis.

## Sample Prep: sensitive phosphoproteomics on single spheroids and monolayer culture of HCT116 cancer cells treated with 5-fluorouracil

Multicellular spheroids and monolayer culture cells were grown from HCT116 cells (ATCC CCL-247). Prior to seeding, cells were harvested from normal cell plates and counted. For spheroids generation, 20,000 cells were seeded on ultra-low attachment 96-well plates (Corning CoStar, Merck) and cultured in 90% DMEM (Gibco, Invitrogen), supplemented with 10% heat-inactivated fetal bovine serum (FBS, Gibco) and 10,000 U/mL penicillin and streptomycin (Invitrogen). Subsequently, spheroids were cultured for 96 h at 37 °C, in a humidified incubator with 5% CO2. Cell medium was refreshed after 48 h, by aspirating half the old medium (making sure not to alter the spheroid) and adding the same amount of fresh medium. For monolayer culture, 20,000 cells were seeded on 24-well plates. After 96 h the spheroids and monolayer-cultured cells were treated with 1.8 µM 5-fluorouracil (Sigma-Aldrich) for 1, 3, 6, 12, and 24 h. Subsequently, the spheroids were harvested by resuspension in 200 µl boiling lysis buffer (5% SDS, 5 mM TCEP, 10 mM CAA, 100 mM Tris pH 8.5) and mixed in a thermo-shaker (1000 rpm) at 95 °C until the entire spheroid disaggregates (-10 min). Monolayer-cultured cells were lysed with 200 µl of boiling lysis buffer (5% SDS; 100 mM Tris pH 8.5, 5 mM TCEP, and 10 mM CAA) and incubated at 95 °C, for 10 min with mixing (1000 rpm). Afterwards, lysates were sonicated with a probe (1 min, 1 second on, 1 second off 50% amplitude, 2 mm Fisherbrand™ Probe for Model 120 Sonic Dismembrator).

Lysates were digested using the Protein Aggregation Capture protocol in the Kingfisher robot modified for low-input amounts. The ratio of MagReSyn® Hydroxyl beads to protein used was 16:1, and the ratio of enzymes used was 1:100 for lysC and 1:50 for trypsin. Samples were digested for 6 h in 200 µl of 50 mM triethylammonium bicarbonate. Digested peptides were acidified after digestion with 50 µl of 10% formic acid. Peptides were concentrated in a SpeedVac at 45 °C until volume was 20 µl. Peptides were resuspended in Loading buffer (80% ACN; 5% TFA, 0.1 M Glycolic Acid) and subjected to phospho-enrichment in the Kingfisher Robot. Process was same as above, but modifying the Loading buffer and WB1 to contain only 0.1 M glycolic acid. 5 µl of MagReSyn® ZrIMAC-HP beads (20 mg/ml) were used per sample, and two sequential enrichment were performed, without changing buffers in between. Samples were eluted in 200 µl of 1% NH3OH and subsequently acidified with 40 µl of 10% TFA. Prior to evotipping, samples were filtered (1 min, 500 × $g$, MultiScreenHTS HV Filter Plate, 0.45 µm, clear, non-sterile). 0.5 µl of the SureQuant™ Multipathway Phosphopeptide Standard (100 fmol/µL) was added to each sample. Finally, samples were loaded into Evotips for subsequent MS analysis.

## Implementation of hybrid-DIA scans on a quadrupole Orbitrap mass spectrometer

The Orbitrap Exploris 480 mass spectrometer was operated with the instrument control software Tune v3.0 or higher (Thermo Fisher Scientific). A standard DIA MS method was built within Xcalibur (v4.3, Thermo Fisher Scientific). Hybrid-DIA scans were customized and programmed via an application interfacing program (API) tool (moonshot_v1.3 or higher) provided by Thermo Fisher Scientific. A full guide on how to operate the API is provided as Supplementary Note 1. Briefly, the hybrid-DIA API works as follows:

1. The theoretical mass-to-charge value, charge state, and retention time window of internal standard (IS) peptides and corresponding endogenous (ENDO) peptides, as well as the theoretical mass-to-charge values of the fragments of IS peptides are predefined as an input.txt file for hybrid-DIA API program. Moreover, the following parameters for the MS2 acquisition are indicated in the API graphic interface: acquisition time, mass tolerance, defined first mass, NCE, isolation width, AGC target, maximum injection time (in milliseconds), MS Trigger Intensity Threshold and dynamic exclusion (in seconds).
2. The precursors of IS peptides are analyzed in MS scans. When IS peptides are detected within the given retention time range, predefined mass tolerance, and above the intensity threshold in MS scan, a fast multiplexed (MSx) PRM MS/MS scan of all detected IS peptides is inserted and performed.
3. When a threshold of predefined fragments for any IS peptide are detected in the PRM MS2 scans within the defined mass tolerance, a multiplexed PRM MS2 scan of the IS peptide and its corresponding endogenous peptide (ENDO) is performed, where the maximal ion injection times for IS and endogenous peptide are set individually to maximize the detection sensitivity for the low abundant endogenous peptide while maintaining a fast DIA cycle time. These co-isolation scans occur for entire list of successfully analyzed IS peptides.
4. If step 2 fails to match the predefined conditions, the mass spectrometry continuously acquires the standard DIA data.
5. Following the completion of all MSx PRM scans of IS peptides and their corresponding ENDO peptide pairs (step 3), the mass spectrometry continuously acquires the standard DIA data. Steps 2 and 3 will repeat whenever the predefined precursors and fragments are identified, respectively.

## LC-MS/MS analysis

Samples were analyzed on the Evosep One system using EV-1112 column (PepSep, 15 cm × 75 μm, beads 1.9 um) and EV-1087 emitter (fused silica, 20 μm). The column temperature was maintained at 40 °C using a butterfly heater (PST-ES-BPH-20, Phoenix S&T) and interfaced online using an EASY-Spray™ source with the Orbitrap Exploris 480 MS (Thermo Fisher Scientific, Bremen, Germany) using Xcalibur (tune version 3.0 or higher). Alternatively, the 5-fluorouracil treated HCT116 samples were analyzed using an IonOpticks Aurora™ column (15 cm-75 μm-C18 1.6um) interfaced with the Orbitrap Exploris 480 MS using a Nanospray Flex™ Ion Source with an integrated column oven (PRSO-V2, Sonation, Biberach, Germany) to maintain the temperature to 50 °C. In all samples, spray voltage was set to 1.8 kV, funnel RF level at 40, and heated capillary temperature at 275 °C. All experiments were acquired using 20 samples per day (SPD) gradient, except for the target dilution and phospho-optimization experiments, which were acquired using 40 SPD.

For full phospho-proteome hybrid-DIA analysis, full MS resolution were set to 120,000 at m/z 200 and full MS AGC target was 300% with an IT of 45 ms. Mass range was set to 350–1400. AGC target value for DIA scans was set at 1000%. Resolution was set to 30,000 and IT to 54 ms and normalized collision energy was 27%. DIA windows scanning from 472 to 1143 m/z with 1 m/z overlap were used (i.e., 11 windows of

61.1 Da for 1 second cycle time at 30 K resolution). To enable non-isochronous injection times for MSx scans, the option must be enabled in Tune (available in Diagnostics > Method Setup).

For hybrid-DIA inclusion lists, the retention time schedule was calculated from Survey Scans runs, where an inclusion list containing the m/z and charge of the spiked-in IS peptides was used to specifically trigger their acquisition. In particular, for the A549 dilution series experiment, as well as for the EGF+ Inhibitors experiment, the retention time schedule was obtained from the SureQuant runs used in those experiments. In both cases, data was imported to Skyline, where the peptide peak integration was manually validated, and the retention times were exported for hybrid-DIA analysis.

Hybrid-DIA acquisition was performed using the moonshot-app (version 1.3 or higher). Parameters for the hybrid-DIA acquisition were: 116 ms of maximum injection time, 10 ppm of mass error, MS intensity threshold of 1e5 and AGC target of 1e6. Dynamic exclusion was set to 5 seconds.

For SureQuant acquisition, we used the template available in Thermo Orbitrap Exploris Series Method Editor. Full-scan mass spectra were collected with a scan range: 300–1,500 m/z, AGC target value: 300%, maximum IT of 50 ms, and 120,000 resolution. Several branches were used, each one for a unique isotopically labeled amino acid and charge state, which will determine the m/z offset. In particular, the method contained 8 branches for +2, +3, and +4 charge states of IS lysine (K8+) and arginine (R10+), as well as +3 charge state of IS alanine (A4+) and +2 charge state of IS valine (V6+) peptides. In each branch, the peptide m/z, charge, and intensity thresholds are defined in the Targeted Mass filter node. For all peptides, intensity threshold was fixed to 1e5. Next, parameters for the fast/survey ddMS2 scans are defined. Resolution was set to 7500 and IT to 10 ms and normalized collision energy was 27%. This is followed by the Targeted Mass Trigger filter node, which defines up to 6 product ions used for pseudo-spectral matching, allowing 10 ppm mass tolerance and minimum of at least 3 detected fragments for each precursor. This step is followed by a sensitive/triggered ddMS2 scan. For the sensitive scan, we used a specific isolation offset for each branch. Resolution was set to 60,000 and IT to 116 ms and normalized collision energy was 27%.

In both acquisition methods, inclusion list for peptides contained in the SureQuant™ Multipathway Phosphorylation Mix was reduced from 131 to 129, due to lack of detection of the precursors of two peptides from that mix (i.e., GSK3:S9 and TSC2:S1387).

## Data analysis: DIA-based discovery pipeline

For hybrid-DIA analysis, DIA scans were extracted using the HTRMS convertor tool from Spectronaut (v15.4 or higher) indicating hybrid-DIA conversion in Conversion type. HTRMS resulting files were further used for directDIA search in Spectronaut (v17). This step is not strictly required since Spectronaut v17, where hybrid-DIA is supported in native Spectronaut analysis.

MS files, both from standard DIA (raw) and hybrid-DIA (HTRMS) were searched using Spectronaut with a library-free approach (directDIA) using a human database (Uniprot reference proteome 2022 release, 20,598 entries). Carbamidomethylation of cysteine was set as a fixed modification, whereas oxidation of methionine, acetylation of protein N-termini and phosphorylation of serine, threonine and tyrosine were set as possible variable modifications. In the HeLa+EGF and spheroids experiment, we filtered out 'b-ions' to prevent quantitative interference from heavy peptides. The maximum number of variable modifications per peptide was limited to 5. PTM localization Filter was checked and PTM localization cutoff was set to 0.75. Cross-run normalization was turned off.

Phosphopeptide quantification data was exported and collapsed to site information using the plugin described in Bekker-Jensen et al.[5] (see Code Availability) in Perseus (v1.6.5.0). Phosphosites intensities were log2-transformed and values were filtered

**Table 1 | Transition settings used in Skyline when importing hybrid-DIA or SureQuant data**

| Transition settings | HybridDIA | SureQuant |
|---|---|---|
| **Full-Scan** | | |
| MS filtering | | |
| Isotope peaks included | None | Count |
| Peaks | NA | 1 |
| Precursor mass analyzer | NA | Orbitrap |
| Resolving power, At | NA | 60,000, 400 mz |
| MS/MS filtering | | |
| Acquisition method | PRM | SureQuant |
| Product mass analyzer | Orbitrap | Centroided |
| Resolving power, At | 30,000, 400 mz | |
| Mass accuracy | | 10 ppm |
| Include all matching scans | TRUE | TRUE |
| **Filter** | | |
| Peptides | | |
| Precursor charges | 2,3,4 | 2,3,4 |
| Ion charges | 1,2,3,4 | 1,2,3,4 |
| Ion types | y | y,b |
| **Instrument** | | |
| Min m/z | 160 | 160 |
| Max m/z | 3000 | 3000 |
| Method match tolerance m/z | 0.001 | 0.001 |
| Triggered chromatogram acquisition | FALSE | TRUE |

to keep only phosphosites quantified in at least 3 replicates in one experimental condition. Data was exported and further processed in R (v4.1.1). Normalization was performed using loess function from limma package (v3.50.3)[56] to correct for loading bias between samples and conditions. Imputation of missing values was performed in two steps using the DAPAR package (v1.26.1)[57] taking into account the nature of the missing values, as described by Lazar et al.[58] First, we considered partially observed values as those values missing within a condition in which there are valid quantitative values in other replicates. These partially observed values were imputed using the slsa function. Secondly, values missing in an entire condition were imputed using the detQuant function from imp4p package (v1.2). Finally, differential expressed phosphosites were calculated using limma (two-sided, BH-FDR < 5%, robust), requiring at least three valid values in one of the two experimental conditions compared.

## Data analysis: targeted pipeline
Raw files acquired in hybrid-DIA mode were processed to extract separately the DIA scans for full phosphoproteome analysis and the IS/ENDO multiplexed scans for targeted analysis.

Multiplexed scans containing the internal standard and the endogenous peptide were extracted in an mzML file using an in-house designed python GUI (python v3.9.7, tkinter v8.6, pathlib v1.0.1) that employs the python library pymsfilereader (https://github.com/frallain/pymsfilereader, v1.0.1) and MSConvert[59,60]. Resulting mzML files were loaded into a Skyline-daily (v21.1.9.353) to extract the intensity information of IS/ENDO scans. Resulting files were used for injection time correction and peak area (AUC) calculation using the R-based (v4.0) shiny-app developed for this purpose. A complete guide to further process the hybrid-DIA scans and perform the IT normalization is available as Supplementary Note 1. The python GUI as well as the shiny-app for IT normalization and visualization are available as the github page for this project: https://github.com/anamdv/HybridDIA.

Skyline template containing the phosphopeptide library of the SureQuant Multipathway Phosphorylation Kit was initially provided by Thermo Fisher Scientific, but then manually curated per experiment to remove shared fragments between isoforms and interfering transients. SureQuant raw files and mzML files from hybrid-DIA runs were imported into Skyline using the abovementioned template using specific transition settings for each acquisition method (Table 1).

Hybrid-DIA quantification was performed using the AUC calculated as mentioned above. Intensities from ENDO peptides were normalized based on the IS peptide intensities. If a significant bias on peptide loading was observed in the DIA data (such in the spheroid dataset), a second normalization step was performed, by using median intensity from DIA scans to correct ENDO peptide intensity.

SureQuant quantification was extracted directly from Skyline, using the data from the Quantification_IS-ENDO report, in particular from the column Ratio To Standard.

## Reporting summary
Further information on research design is available in the Nature Portfolio Reporting Summary linked to this article.

## Data availability
The mass spectrometry proteomics data have been deposited to the ProteomeXchange Consortium via the PRIDE[61] partner repository with the dataset identifier "PXD038947". Uniprot reference proteome used in Spectronaut searches was 2022 release (20,598 entries). Source data are provided with this paper.

## Code availability
Custom Python and R code used in the manuscript is available in the GitHub repository https://github.com/anamdv/HybridDIA. Hybrid-DIA API can be downloaded from https://github.com/thermofisherlsms/MoonshotApps.

PTM collapse plugin requires Perseus and R (minimum version 3.6.0) to run and it is available at https://github.com/AlexHgO/Perseus_Plugin_Peptide_Collapse.

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

## Acknowledgements

The project was part of a research collaboration 'cSHARP' between Thermo Fisher Scientific, University of Copenhagen, Kyoto University and Academia Sinica. Work at The Novo Nordisk Foundation Center for Protein Research (CPR) is funded in part by a generous donation from the Novo Nordisk Foundation (NNF14CC0001). This work has also been funded as part of EPIC-XS project under the grant agreement no. 823839 funded by the Horizon 2020 programme of the European Union and supported by the European Research Council through ERC-Synergy grant 810057-HighResCells. C.K. is supported by the Marie Skłodowska Curie European Training Network "PUSHH" (grant number No. 861389). We thank Aaron Gajadhar and Bhavin Patel from Thermo Fisher Scientific for providing us early access to the SureQuant™ Multi-pathway Phosphopeptide Standard kit.

## Author contributions

A.M-V. designed the experiments, optimized the API usage, performed all proteomics experiments, analyzed the data, and implemented the data analysis pipeline. C.K. and G.F. helped in the experimental design and optimization of the phosphorylation enrichment protocol. L.V.H. and G.F. generated the spheroids model. L.V.H. prepared the samples in the 5-FU treatment experiment. K.L.F., Y.X., T.M., and A.A.M. contributed to the development of the API. Y.C. and Y.I. provided input on experiments and evaluated results. J.V.O. designed the experiments and critically evaluated the results. A.M.V., Y.X., and J.V.O. wrote the manuscript. All authors read, edited, and approved the final version of the manuscript.

## Competing interests

The authors declare the following competing financial interest(s): K.L.F., Y.X, T.M., and A.A.M. are employees of Thermo Fisher Scientific, the manufacturer of the Orbitrap Exploris 480 MS instrument used in this research. Thermo Fisher Scientific provides support to J.V.O.'s laboratory under a confidentiality agreement with Novo Nordisk Foundation Center for Protein Research, University of Copenhagen. The intelligent data acquisition Hybrid-DIA MS as a methodology of mass spectrometry is the subject of a patent application filed in 2019 and published in 2021 in multiple countries, as CN112798696A; DE102020129645A1; GB2590601A; US2021225627A1. The patent application can be found at: https://worldwide.espacenet.com/patent/search/family/068988062/publication/GB2590601A?q=B2590601A. The remaining authors declare no competing interests.
