## [Peer Review File · Nature Communications]

REVIEWER COMMENTS

Reviewer #1 (Remarks to the Author):

Review of Ana Martinez-Val et al. Hybrid-DIA

MS-based, bottom up proteomics has recently pivoted mostly from data dependent acquisition (DDA) to data independent acquisition (DIA). However, there is also 'targeted acquisition', technically a sub-category of DDA, which is used in routine application as opposed to discovery applications. In this interesting manuscript, the authors bring together the world of DIA with the targeting world. This is made possible by an application programming interface (API) that the instrument manufacturer (Thermo Fisher Scientific) provides. They are also co-authors of this paper, together with three academic groups.

To my knowledge, this is the first paper ever to combine DIA with targeted acquisition in a hybrid acquisition method in real time. I imagine a great future for this approach in applications such as the ones the authors describe and many more. Thus, the magnitude of the advance as well as the general importance of it, warrant publication in Nature Communications in my opinion.

The manuscript itself is generally well written but still needs work. There are numerous occasions where the text should be improved ranging from grammatical mistakes to style ('way-to-go method' in the abstract, for example). Furthermore, the overall logic and flow could also be improved.

In the title and in the text, it is not quite clear to me if the authors claim to be the first with this hybrid idea (they should also mention patents that they or at least Thermo has presumably filed on this method). So if this is the case, the title is unnecessarily restrictive and if this is not the case the authors should make that clear early on in the paper. Presumably, the focus on phosphoproteomics is mainly to have a specific use case, and this should be made clear in the introduction.

At the beginning of the results, I am missing some background on how the spiked in peptides are detected and used for triggering. The paper just says 'inclusion list', but is any intelligence involved like in MaxQuant.Live that presumably uses the same API but without the aid of standards? The separation of the targeted results from the DIA results and funneling into Skyline appear cumbersome. Are there efforts to streamline this?

The first example is a proof of principle, using synthetic peptides from Pierce. This example and Figure 1 do a good job of explaining the method and the linearity shown is excellent, but it would be good if the authors quantified the theoretical gain in sensitivity some more. It is probably the length of the purple box compared to the length of the green boxes, but what factor is this? Are we restricted to one endogenous peptide per DIA cycle and why?

The remainder of the paper deals with phosphoproteomics, with the authors showing an impressive 2.4 ug starting material experiment where the targeted phosphopeptides were still detected. After having compared their hybrid DIA method to sure-quant in DIA, they next compare it to the sure-quant targeted method using the EGFR pathway and conclude that the results for the targeted peptides are

quite comparable, with hybrid-DIA also providing unbiased phospho-data on pathways missed by the sure-quant panel.

in an organoid system. In these experiments, they also introduce an improved, high sensitivity protocol. Application of all this to chemotherapy (5-FU) treatment of the organoids resulted in interesting new insights.

Reviewer #2 (Remarks to the Author):

Having the ability to measure (e.g., phosphorylation) globally from smaller sample sizes makes it possible to investigate more questions and/or conditions with higher resolution (time and/or concentration). The global aspect is crucial in the discovery phase of a research project in order to determine which parameters best characterize the research question at hand. According to my assessment, Martínez-Val et al. et al do a good job anchoring the new method in relation to existing alternatives and demonstrating the feasibility of acquiring comprehensive information from smaller samples. Overall I am impressed by the study and I believe that that the presented method/technology will enable researchers to generate more data from experiments on relevant tissues/models (e.g. organoids/spheroids).

Specific questions:

1. What was the basis for the choice of cell lines? Why are these good starting points for the scientific questions asked? It is unclear in the main text which cell type was used (for the 2D/3D experiments) and why (although it is possible to read about it in materials and methods and figure legend). In my opinion, describing the rationale and context makes it more interesting to read about the technical improvements.

2. Which method of cell line authentication has been utilized?

3. The 5-FU concentration (1.8 μM) appears to be quite low. How was it chosen and have any supporting viability analyzes been performed to support that this particular concentration was chosen? Viability or similar would set the observed effects in a relevant perspective.

4. What was the reasoning for the choice of a hydrophilic compound 5-FU when analyzing effects in a 3D model? The penetration of 5-FU is known to be poor, and it is not an ideal compound if you would like

to study effects deep in the parenchyma. In this context it is worth mentioning that a spheroid culture with medium change results in a mixed model with proliferating cells towards the spheroid surface.

5. What was the volume of medium per well in the 24-well plate vs the 96-well plat? Given that it's the same number of are cells seeded, could this also affect the microenvironment (more than 2D vs 3D growth)?

6. What was the vehicle for 5-FU? DMSO? Was any vehicle control analyzed?

7. As far as I can tell, the temporal changes are compared to time point 0. What about vehicle controls for the respective time points? How much difference do you observe in untreated controls over this time (1, 3, 6, 12 and 24 hrs)? If you observe significant difference over this period, the more relevant comparison would be treatment against each time matched vehicle treated control, to isolate the treatment specific changes.

8. When comparing 2D and 3D cultures (untreated), do you observe any signs of expected biological differences like altered cell cycle distribution and induced HIF1a mediated hypoxic signaling?

Minor point

9. Regarding figure 3, I would prefer if the inhibitors were named in the figure legend to more easily see which inhibitors were use.

Reviewer #3 (Remarks to the Author):

It is an interesting idea to combine DIA and targeted proteomics in a single shot. Since DIA data can be inherently analyzed by targeted data analysis, sacrificing some of the scanning time (therefore fewer peptides could be analyzable) for targeted assays might be necessary if the sensitivity and reproducibility could be significantly improved. The key part of this Hybrid-DIA method is the benchmarking against the standard DIA method and PRM method. Ideally, readers should expect this hybrid method is more sensitive than DIA and more comprehensive than PRM. However, the manuscript did not present sufficient data to support this aspect.

The major benefits of Hybrid-DIA should be shown in Figure 2. B shows only three phosphopeptides but the authors did not explain why these three were chosen. One might expect to see the overview of all the 131 peptides. More importantly, the benefit of Hybrid-DIA is unclear from Fig2B. Although there are more interfering traces in the DIA, the peak groups in DIA look not too bad. The authors should quantitatively present the benefits of Hybrid-DIA in a systematical way.

Fig2E shows lower correlation between DIA and hybrid-DIA when the sample injection amount decreases, but which is closer to the ground truth? Reader might expect hybrid-DIA is more quantitatively precise for some peptides with targeted analysis, but how about the others? The authors should largely expand the analysis of this section so that readers could obtain a more comprehensive view of the benefits (and limitations, if any) of hybrid-DIA.

In Fig3, the authors showed hybrid-DIA generated more data than SureQuant. Again, the authors did not show systematically and quantitatively how much % of accuracy was improved or sacrificed using hybrid-DIA compared to SureQuant. They showed some profiles of selected phosphopeptides based on a signaling pathway, however, for this methodology paper, more analytical analyses are recommended.

Minor issues:

- 1.It would be better to briefly clarify the importance of comprehensively quantification of phosphoproteomics in biological or clinical samples in the introduction.
- 2.Background for library free phosphoproteomics on page 6 should be put in the introduction.
- 3.Please add more legends or text sign in the Figure 1A to help understand the acquisition strategy, e.g., the red and purple rectangles for PRM and MSx, respectively. Besides, if the DIA data are simultaneously acquired with PRM and MSx as described on page 5, the green rectangles standing for DIA windows should not be put behind the red and purple rectangles denoting PRM and MSx.
- 4.The reviewer advises changing Figure 2D from the current line plot to histogram, and labelling the phosphor-site identification numbers for clarity.
- 5.The reviewer advises adding some statistical test in Figure 3C.
- 6.What does the activity in Figure 3D mean and how was it assessed?
There is no annotation for the yellow group in Figure 3E or its legend. I guess yellow is for the group without any inhibitor? Please clarify.
- 7.Please specify in Figure 4C which panel belongs the results of spheroid or monolayer. It is better to display the figure in the form of split violin plot.
- 8.There are several typos and inappropriate expressions. Some examples were listed as following:

Page 3: Change “required” to “require”.

Figure 4B: Change “Spheroids” to “Spheroid”.

Point-by-point response rebuttal letter

NCOMMS-23-03598-T

Our answers to the reviewer questions are indicated in blue text. New additions or modifications in the manuscript are highlighted in yellow.

Reviewer #1 (Remarks to the Author):

Review of Ana Martinez-Val et al. Hybrid-DIA.

MS-based, bottom up proteomics has recently pivoted mostly from data dependent acquisition (DDA) to data independent acquisition (DIA). However, there is also 'targeted acquisition', technically a sub-category of DDA, which is used in routine application as opposed to discovery applications. In this interesting manuscript, the authors bring together the world of DIA with the targeting world. This is made possible by an application programming interface (API) that the instrument manufacturer (Thermo Fisher Scientific) provides. They are also co-authors of this paper, together with three academic groups.

To my knowledge, this is the first paper ever to combine DIA with targeted acquisition in a hybrid acquisition method in real time. I imagine a great future for this approach in applications such as the ones the authors describe and many more. Thus, the magnitude of the advance as well as the general importance of it, warrant publication in Nature Communications in my opinion. The manuscript itself is generally well written but still needs work. There are numerous occasions where the text should be improved ranging from grammatical mistakes to style ('way-to-go method' in the abstract, for example). Furthermore, the overall logic and flow could also be improved. In the title and in the text, it is not quite clear to me if the authors claim to be the first with this hybrid idea (they should also mention patents that they or at least Thermo has presumably filed on this method). So if this is the case, the title is unnecessarily restrictive and if this is not the case the authors should make that clear early on in the paper. Presumably, the focus on phosphoproteomics is mainly to have a specific use case, and this should be made clear in the introduction.

We appreciate the reviewer feedback and following their suggestion we have resubmitted a revised version in the manuscript in which we expect to have improve the text, logic and flow thanks to re-writing key sections and providing new data to facilitate understanding the methodology and logic behind the hybrid-DIA method.

Furthermore, we have included a statement about the patent files by Thermo (lines 763-768) that described this novel method. Moreover, we have taken into consideration the suggestion of this reviewer in regards to the novelty of the technique, so we have rephrased the abstract (line 15-28) to emphasize that it is the first time an acquisition method like this one is described.

Finally, we have modified the introduction as suggested to indicate that the phospho-proteomics examples is a specific use case (line 101-106). In this line, we have update the title: "Hybrid-DIA: Intelligent Data Acquisition for Online Targeted and Discovery Proteomics Applied to Phosphoprotein signaling in Single Spheroids".

At the beginning of the results, I am missing some background on how the spiked in peptides are detected and used for triggering. The paper just says 'inclusion list', but is any intelligence involved like in MaxQuant.Live that presumably uses the same API but without the aid of standards?

We apologize for not providing enough information about the detection of the spiked in peptides. We have now refer in the main text (line 121, line 132) to the method section –Supplementary Data 1 and Method section “Implementation of hybrid-DIA scans on a quadrupole Orbitrap mass spectrometer” (lines 762-799)- where the logic of the hybrid-DIA API is explained.

Moreover, we have included some examples of how the spiked-in heavy labelled standards (IS) are detected and the multiplex scan with their endogenous counterpart (MSx IS/ENDO) is triggered (Supplementary Figure 1).

We also included a better explanation of how the inclusion list must be created in the main text (lines 128-132). In brief, the inclusion list must contain the fragments of the spiked-in peptides that will be detected by the API to trigger the MSx IS/ENDO scan. This inclusion list needs to be generated prior to the hybrid-DIA analysis, by performing a targeted analysis of the precursor masses of the spiked-in peptides. From that analysis, we can determine the top-n fragments to use for subsequent detection, as well as the expected retention time. In the case of the SureQuant Multipathway Phosphorylation Mix such initial template (top 6 fragments per peptide) was provided by Thermo Fisher Scientific, and the retention time was obtained from several controls runs containing the heavy spiked-in peptides in a complex matrix (HeLa phospho-enriched samples), which were acquired using the same gradient and set-up as the samples.

Also, we have improved the description of the scheme in figure 1A where we show graphically how hybrid-DIA run looks like. Briefly, during a hybrid-DIA run, the API would look for the mass of the spiked-in heavy peptide in the full scans acquired in the time window specified in the inclusion list. Those spiked-in heavy peptides detected within a certain error mass would be co-fragmented in one scan. Then, the API would look in that scan for the fragment masses provided in the inclusion list. If at least 3 fragments are detected with a certain Signal-to-noise level, then the MSx scan of that spiked-in peptide with its endogenous peptide will the created in the same cycle.

The separation of the targeted results from the DIA results and funneling into Skyline appear cumbersome. Are there efforts to streamline this?

We understand that the pipeline to analyze hybrid-DIA data could seem cumbersome, due to the multiple steps required to achieve the final quantification matrix. However, our pipeline allows the user to process the targeted data using freely available software (all third party software for the targeted analysis is freely available at its corresponding sources, as indicated in Supplementary Data 2).

However, to facilitate the user experience, we have been in contact with Biognosys and they have implemented hybrid-DIA support in SpectroDive (v11). SpectroDive (v11) can directly read the .raw files and identify that they proceed from a hybrid-DIA experiment, and provided the appropriate panel (or library of heavy peptides), it can easily do the downstream injection time normalization and endogenous to heavy ratio. We have included this in the main text (lines 158-159 and lines 243-246) as well as in Supplementary Figure 2C-D in the revised manuscript (also below as Rebuttal Letter Figure 1A-B).

Regarding the possibility to streamline this process even further so as to process the DIA and targeted data in one-round of analysis, we have not envisioned that for the current project, but surely it would be very interesting in the future updates of this application.

Rebuttal Letter Figure 1:

The first example is a proof of principle, using synthetic peptides from Pierce. This example and Figure 1 do a good job of explaining the method and the linearity shown is excellent, but it would be good if the authors quantified the theoretical gain in sensitivity some more. It is probably the length of the purple boxes compared to the length of the green boxes, but what factor is this? Are we restricted to one endogenous peptide per DIA cycle and why?

Figure 1 is just for explanatory purposes of the method, so it was designed as a graphical example of a run using hybrid-DIA the method. It was design for pedagogical purposes, and therefore, the boxes are not set to scale. To improve this explanation in the main manuscript, we have included a more thorough description of the figure 1A legend and in the Main text (lines 128-143). Further details of the logic of the API algorithm are provided as Supplementary Data 1.

Importantly, to answer the question about number of endogenous peptides per DIA cycle: there is no restriction on the number of endogenous peptides that can be triggered per DIA cycle. In our data, up to nine peptides can be triggered in the same DIA cycle (see example below in Rebuttal Letter Figure 2 and 3). In the first example (Rebuttal letter figure 2), nine IS peptides where detected in the full scan, and therefore the API created a scan where fragments from all nine peptides where analysed simultaneously (panel A). The API will then analyze those fragments and match them with the ones provided in the inclusion list. For those peptides that have at least three fragments identified, the API will create the corresponding MSx IS/ENDO scans in the same cycle. In this example, 6 out of 9 peptides were identified. Panel B shows the matching y-series of the identified peptides using the “Interactive Peptide Spectral Annotator” from Brademan et al (PMID: 31088857). Finally, in panel C, the generated MSx IS/ENDO scans are plotted. In summary, in this example, the full cycle was comprised of one full scan, followed by 11 DIA windows, 1 IS multiplexed scan and 6 IS/ENDO multiplex scans. The next example, from the same MS run, shows the same but in this case 6 IS peptide were found in the full scan (panel A), and 5 of them were identified (panel B), therefore triggering the corresponding IS/ENDO MSx scans (panel C) in that cycle.

Multiplexing co-eluting IS is crucial for the performance of this tool, since it allows to minimize the duty cycle time used to scan for those peptides. We have highlighted this in the revised text (lines 133-134, Supplementary Figure 1).

Rebuttal Letter Figure 2:

A

B

C

Rebuttal letter Figure 3:

A

20220901_EXPL2_EVO7_AMV_SA_40SPD_hyDIA_Phos_HeLa_30k1s_IonOptics_TiMAC_10ug_100targets_01

B

C

On the other hand, we agree with this reviewer that we should show better the quantitative gain in sensitivity of the MSx scans when compared to standard DIA scans. To do so, we have included an example that shows the actual gain for a low abundant peptide species (shown below as Rebuttal letter figure 4). Here, we have analyzed the proteome of A549, a lung adenocarcinoma cell line that carries a mutation on KRAS at position 12 (Gly → Ser). For that purpose, we spiked in heavy-labelled KRAS-G12S peptide into 100 ng A549 peptide digests and then acquired the sample using hybrid-DIA.

As it can be seen in panel A below, the fragments y7, y9 and y9 of the heavy-labelled KRAS-G12S can be clearly seen in the corresponding DIA window. On the other, the fragments of the endogenous KRAS-G12S peptide are very low (and not clearly visible) in this DIA spectrum, since they are very close to the noise. However, when we look at the MSx scan triggered by the hybrid-DIA API in the same acquisition cycle, the signal of the fragments y7, y8 and y9 from the endogenous peptide can be clearly observed, even though the fragments from the heavy-labelled peptides are still dominant. This is achieved thanks to the use of a non-synchronous injection time during the MSx scan that allows the instrument to accumulate the endogenous peptide for 116 milliseconds, while it only accumulate the heavy-labelled peptides for 13 milliseconds. In contrast, due to the complexity of the DIA window where the endogenous peptide would be contained, the overall injection time is of 2.5 milliseconds.

To show this win in more systematic and quantitative manner, we measured the signal-to-noise (fragment raw intensity divided by the noise) of each fragment in the DIA window and the immediate following MSx scan, in three independent replicates (panel B below). It is clear from the plot in panel B, that the signal-to-noise for the endogenous peptide is significantly boosted when acquired using hybrid-DIA triggered MSx scans.

This new analysis is shown in the revised manuscript in lines 161 to 171 and Figure 1C.

Rebuttal Letter Figure 4:

The remainder of the paper deals with phosphoproteomics, with the authors showing an impressive 2.4 ug starting material experiment where the targeted phosphopeptides were still detected. After having compared their hybrid DIA method to sure-quant in DIA, they next compare it to the sure-quant targeted method using the EGFR pathway and conclude that the results for the targeted peptides are quite comparable, with hybrid-DIA also providing unbiased phospho-data on pathways missed by the sure-quant panel. In an organoid system. In these experiments, they also introduce an

improved, high sensitivity protocol. Application of all this to chemotherapy (5-FU) treatment of the organoids resulted in interesting new insights.

We appreciate the positive feedback on the phospho-proteomics experiments and applications shown in the manuscript.

Reviewer #2 (Remarks to the Author):

Having the ability to measure (e.g., phosphorylation) globally from smaller sample sizes makes it possible to investigate more questions and/or conditions with higher resolution (time and/or concentration). The global aspect is crucial in the discovery phase of a research project in order to determine which parameters best characterize the research question at hand. According to my assessment, Martínez-Val et al. et al do a good job anchoring the new method in relation to existing alternatives and demonstrating the feasibility of acquiring comprehensive information from smaller samples. Overall I am impressed by the study and I believe that that the presented method/technology will enable researchers to generate more data from experiments on relevant tissues/models (e.g. organoids/spheroids).

We appreciate the positive feedback of this reviewer when pointing out the relevance of the presented methodology to advance the research on relevant tissue models.

Specific questions:

1. What was the basis for the choice of cell lines? Why are these good starting points for the scientific questions asked? It is unclear in the main text which cell type was used (for the 2D/3D experiments) and why (although it is possible to read about it in materials and methods and figure legend). In my opinion, describing the rationale and context makes it more interesting to read about the technical improvements.

We apologize for not being clearer in this regard in the first version of the manuscript.

On the one hand, for our initial method optimization experiment where we performed phospho-enrichment of decreasing amounts of peptides we chose A549 cell line. A549 cell line has been previously used in our research group (PMID: 31585087) and we consider it a good model for phospho-proteomics and general proteomics optimization analysis. On the other hand, we chose HeLa for EGF stimulation experiment because HeLa is a good model to study EGFR downstream signaling, since it expressed moderate levels of EGFR and it is suitable for proteomics and signaling studies, as we have proven in multiple publications (PMID: 34876567, PMID: 34876567, PMID: 17081983).

On the other hand, for our last experiment where we studied the difference of drug treatment in 2D vs 3D cell culture we chose HCT116 since this cell line can form spheroids easily and there is multiple published evidence of using HCT116 as a model for 3D cell culture (PMID: 27198560, PMID: 28650662, PMID: 30061712), as well as comparisons of this cell line growth in 2D vs 3D that we used as a base to design our experiments (PMID: 27313779).

Moreover, we are interested in studying HCT116 derived spheroids as a proxy to study colorectal cancer. Among different colorectal cell lines, HCT116 is one that shows more sensitivity to 5-fluorouracil used in this study, for instance when compared to HT29, which is very resistant to this treatment in spheroid form (PMID: 29141026). Moreover, HCT116 has been classified as CMS4 subtype (PMID: 28683746), which is the most aggressive subtype and the one with worse prognosis.

We have included a brief explanation of the choice of HCT116 cell line in the revised manuscript (lines 471-478).

2. Which method of cell line authentication has been utilized?

HCT116 cells were bought from ATCC for this study. HeLa and A549 were authenticated by STR profiling using the ATCC authentication service according to manufacturers' instructions.

3. The 5-FU concentration (1.8 μ M) appears to be quite low. How was it chosen and have any supporting viability analyses been performed to support that this particular concentration was chosen? Viability or similar would set the observed effects in a relevant perspective.

The 5-FU concentration was chosen based on published data (PMID: 29141026) where the effect of 5-FU was assessed in both 2D and 3D models. In that publication of reference, Virgone-Carlotta et al calculated the IC50 at 48 hours for HCT116 and 5-FU in 2D and 3D grown cells. The authors found that the IC50 for 2D cells was between 2-5 μ M, whilst it was between 1-5 μ M in 3D cells. Based on this data, we chose 1.8 μ M as the lowest value to ensure a signaling effect whilst avoiding cells death during the course of the treatment.

4. What was the reasoning for the choice of a hydrophilic compound 5-FU when analyzing effects in a 3D model? The penetration of 5-FU is known to be poor, and it is not an ideal compound if you would like to study effects deep in the parenchyma. In this context it is worth mentioning that a spheroid culture with medium change results in a mixed model with proliferating cells towards the spheroid surface.

The purpose of this work was to show that hybrid-DIA provided the depth, sensitivity, precision and accuracy required to perform high-throughput (phospho)-proteomics drug screening. We used 3D vs 2D model comparison as a proof-of-concept since the use of spheroids in proteomics has been severely limited due to the challenges of working with this model (i.e., limitation of sample amount). Our first aim was to show that 3D models are more relevant and provide significantly different information than 2D models, and therefore are more significant for drug-screening studies. Secondly, we wanted to show that sample limitation could be overcome by employing our sensitive phosphor-enrichment methodology combined with hybrid-DIA acquisition.

Studying the effect of the drug penetration in the parenchyma was not our main purpose, but we agree with the reviewer that such effects will be limited and should be accounted. These effects would be smaller due to the poor penetration of the compound. Importantly, the signals measured by MS would be a mixture of the signal on the surface versus the signal on the apoptotic core of the spheroid. We have modified the text to make such limitation evident to the reader (lines 610-615).

Finally, we chose 5-Fluorouracil (5-FU) due to its clinical relevance on colorectal cancer (CRC), which our HCT116 cell line was used as model. This drug remains the first-line treatment for colorectal cancer (CRC) being a fundamental component of chemotherapeutic agents for palliative and adjuvant treatments of this type of cancer (PMID: 21180512, 22149436). Moreover, 5-FU has been previously used in the context of comparison 3D vs 2D effect (PMID: 29141026), showing significant effect of the cell model in the effect of the drug, and, most importantly showing that HCT-116 is still sensitive to 5-FU in 3D models. We have clarified this rationale in the revised main text (line 471-478).

5. What was the volume of medium per well in the 24-well plate vs the 96-well plate? Given that it's the same number of cells seeded, could this also affect the microenvironment (more than 2D vs 3D growth)?

The reviewer raises an interesting point regarding the effect of the cell media volume and its impact on microenvironment signaling. In our experiment, the volumes used for the 96 well plate (3D) was 250 μ l, conversely the volume used for the 24-well plate (2D) was 1 ml.

Nakamura et al. (PMID: 33806404) described the role that medium plays in determining the number of proteins detected after growing cells in different medium volumes. It should be noted that this experiment was performed in albumin depleted serum. Here they show that with decreasing volume (in 10 cm dishes) the number of detected proteins increases up to 3x. It is however likely that this phenomena occurs due to change in dynamic range, which can drastically improve the detection of peptides in the mass spectrometer. Additionally, the overlapping proteins remain the same regardless of volume, suggesting that the proteome is mostly unchanged and only the dynamic range changes.

Regarding the micro-environment, the difference in medium volume will definitely affect the cellular secretome concentration and several other factors including the following: nutrient supply, dilution, or concentration of waste products and metabolites, and changes in oxygen level (PMID: 30081523). For some cell lines this can and will affect the expression of genes, and thus the proteome as well (PMID: 28535008). The most well studied example of this is in osteoblasts and osteoclasts where it is shown that the medium volume affects the most important factors for bone homeostasis up to 3x (PMID: 28535008). However, difference between 2D and 3D secretome is significantly more affected, with many inflammatory proteins having difference in secretion of 50x (PMID: 29343288)

So while we do think that some small effects might be caused by the difference in medium culture, and subsequent concentration of waste products, we do not believe this will cause a significant difference, since we assume that the main signaling observed in our data would be due to the addition of 5-FU, and in that case the concentration was constant in both conditions. .

6. What was the vehicle for 5-FU? DMSO? Was any vehicle control analyzed?

5-Fluorouracil was initially suspended into DMSO, and then diluted in milliQ water to achieve the final concentration for cell treatment. The concentration of DMSO in the treated cells was <0.005%. We did not perform any vehicle control analysis, but we acknowledge that it would have been relevant. However, in the end, our purpose was to compare the effect of 2D vs 3D, and since the time 0 and the vehicle during treatment was the same the confounding effects of the vehicle should be compensated.

7. As far as I can tell, the temporal changes are compared to time point 0. What about vehicle controls for the respective time points? How much difference do you observe in untreated controls over this time (1, 3, 6, 12 and 24 hrs)? If you observe significant difference over this period, the more relevant comparison would be treatment against each time matched vehicle treated control, to isolate the treatment specific changes.

We agree with the reviewer that vehicle can affect the signaling pathways. However, in this experiment, as also stated in the previous answer, since our purpose was to compare the same time points in 2D and 3, and since the vehicle was the same in both conditions, we did not include separate vehicle controls.

8. When comparing 2D and 3D cultures (untreated), do you observe any signs of expected biological differences like altered cell cycle distribution and induced HIF1a mediated hypoxic signaling?

In our initial analysis, we already pointed out in Figure 8B that H2AX serine 140, a marker of DNA damage and apoptosis was higher in time zero in spheroids than in adherent cells. However, to expand the knowledge from this work and explain better the initial divergence between the two models compared in this work, we analyzed the differential regulation of the phospho-proteomes at time 0 (Rebuttal Letter Figure 5). As expected, and suggested by this reviewer, there are in fact clear signals of altered cell cycle in 3D when compared to the 2D counterparts. One clear example of this is the

increased phosphorylation of protein RB1, such as threonine 373 (panel A-B below), which is a known substrate of CDK2, and it is required for RB1 inactivation due to disruption of interaction with E2F1 during cell proliferation (<https://doi.org/10.1074/jbc.M110.108167>, PMID: 9447971). Moreover, we found a clear signature of rps6 phosphorylation in adherent cells when compared to spheroids. In fact, rps6 phosphorylation has been described to be lower in 3D vs 2D models (panel A-B, below), due to a negative gradient from surface to the center of the spheroid (PMID: 27663511).

On the other hand, we found that KRT8 was significantly more represented on spheroids than adherent cells. In particular, we observed increased phosphorylation on Serine 74 (panel A-B, below), which is annotated as a JNK substrate (PMID: 11781324). Moreover, KRT8 has already been described as a differentially regulated protein in spheroids when compared with same cells in 2D culture (<https://doi.org/10.1371/journal.pone.0135426>).

Unfortunately, in this dataset we did not find mass-spectrometry based data to map proteins related to hypoxic signaling. This could be due to lack of depth in the analysis, or because of the heterogeneity in the cells population and metabolism, since there is a hypoxic gradient from the surface to the center of them. This heterogeneity can make signals to be diluted in the MS data. We have expanded about this limitation in the discussion, and suggested some interesting works that study spatial heterogeneity in spheroids using spatial SILAC, to differentially labelled the inner core, middle and outer layer (PMID: 34813286), or by serial trypsinization (PMID: 7471046).

We have included this new analysis as finding in the revised manuscript (lines 511-514 and Supplementary Figure 6A).

Rebuttal Letter Figure 5:

Minor point

9. Regarding figure 3, I would prefer if the inhibitors were named in the figure legend to more easily see which inhibitors were use.

We have corrected the figure legend.

Reviewer #3 (Remarks to the Author):

It is an interesting idea to combine DIA and targeted proteomics in a single shot. Since DIA data can be inherently analyzed by targeted data analysis, sacrificing some of the scanning time (therefore fewer peptides could be analyzable) for targeted assays might be necessary if the sensitivity and reproducibility could be significantly improved. The key part of this Hybrid-DIA method is the benchmarking against the standard DIA method and PRM method. Ideally, readers should expect this hybrid method is more sensitive than DIA and more comprehensive than PRM. However, the manuscript did not present sufficient data to support this aspect.

The major benefits of Hybrid-DIA should be shown in Figure 2. B shows only three phosphopeptides but the authors did not explain why these three were chosen. One might expect to see the overview of all the 131 peptides. More importantly, the benefit of Hybrid-DIA is unclear from Fig2B. Although there are more interfering traces in the DIA, the peak groups in DIA look not too bad. The authors should quantitatively present the benefits of Hybrid-DIA in a systematical way.

We understand the concern of this reviewer, so we have decided to perform a systematic analysis of the sensitivity, accuracy and precision obtained in the targeted peptides when using hybrid-DIA and compared it to the results of using standard DIA.

First, to show an example of the significant win on sensitivity when doing the targeted acquisition of pre-defined set of peptides we have included a new panel in figure 1 (shown below). In here, we have analysed the proteome of A549, a cell line that it is known to carry a mutation on KRAS at position 12 that change glycine for serine. This mutation is difficult to observe in a shot-gun experiment when the sample is limited, so we used it as an example of the increased sensitive achieved when using hybrid-DIA. For that purpose, we spiked in heavy-labelled KRAS-G12S peptide into 100 ng A549 peptide digests and then acquired the sample using hybrid-DIA.

Rebuttal Letter Figure 6:

As it can be seen in panel A above, the fragments y7, y9 and y9 of the heavy-labelled KRAS-G12S can be clearly seen in the corresponding DIA window. On the other, the fragments of the endogenous KRAS-G12S peptide are hidden in this DIA spectrum, since they are very close to the noise. However, when we look at the MSx scan triggered by the hybrid-DIA API in the same acquisition cycle, the signal

of the fragments y7, y8 and y9 from the endogenous peptide can be clearly observed, even though the fragments from the heavy-labelled peptides are still dominant. This is achieved thanks to the use of a non-synchronous injection time during the MSx scan that allows the instrument to accumulate the endogenous peptide for 116 milliseconds, while it only accumulates the heavy-labelled peptides for 13 milliseconds. In contrast, due to the complexity of the DIA window where the endogenous peptide would be contained, the overall injection time is of 2.5 milliseconds.

To show this win in a more systematic and quantitative manner, we measured the signal-to-noise (fragment raw intensity divided by the noise) of each fragment in the DIA window and the immediate following MSx scan, in three independent replicates (panel B above). It is clear from the plot in panel B, that the signal-to-noise for the endogenous peptide is significantly boosted when acquired using hybrid-DIA triggered MSx scans.

We have included this new analysis in the revised manuscript (lines 161-171) and Figure 1C.

Furthermore, to improve the benchmark and systematic comparison of hybrid-DIA against DIA, we have extended our initial analysis in the revised figure 3 (rebuttal letter figure 7). In this new analysis, we have provided a general overview of all the targeted peptides monitored in this experiment by plotting a heatmap showing relative quantification (z-score of endogenous versus heavy peptide ratio across samples) of targeted peptides in a dilution series experiment in hybrid-DIA (blue) and DIA (green) (panel A below in rebuttal figure 7). Here, the peptides monitored using the heavy-labelled SureQuant Multipathway Phosphopeptide Kit are ranked by the relative abundance, from peptides from abundant in the sample to those lowest. From this first analysis, it is clear that hybrid-DIA provides more coverage of the targeted peptides even to those less abundant in the sample. On the other hand, for high abundant peptides, DIA provides good quantification, but as soon as the abundance of the phosphopeptide decreases, the signal is lost or extremely noisy. On the contrary, there are some peptides not measured in hybrid-DIA that are measured in DIA. This could be attributed to faults in the scheduling when using the API. However, as shown in the data, these cases are the exception, and potentially, in those cases one can complete the information using the data from the DIA windows in hybrid-DIA mode.

The reviewer also asked about the selection of the three phosphopeptides used to plot the XICs in panel B in former figure 2. We choose them based on their relative abundance in the sample, where AKT1S1:T246 is a very abundant phosphopeptide, and there it is easily quantified in both hybrid-DIA and standard-DIA mode, whilst TSC2:S939 is much less abundant and PLGC1:Y783 is one of the lowest peptides detected. To clarify this selection, which was done for illustration purposes, we indicated in the heatmap with black arrows, the position of these phosphopeptides in the overall rank: 1: AKT1S1-T246; 2: TSC2:S939; 3: PLGC1-Y783 (see panel A, rebuttal figure 7, and revised Figure 3B-C).

Next, we explored the quality of the data obtained for the monitored/targeted peptides in DIA and hybrid-DIA. For that purpose we included a barplot to show the number of targeted/monitored peptides found in each method after applying different quality filters (panel B rebuttal figure 7, and revised Figure 3D). In this barplot, in blue, it is the total number of monitored/targeted peptides with traces detected in Skyline in each method (DIA:top, hybrid-DIA: bottom). Next, in yellow, number of heavy-endogenous pairs detected with a "peptide peak found ratio" or PPF ratio higher than 0.5. PPF ratio indicates the proportion of transition for which Skyline determines that a peak exists and it is co-eluting with the primary peak. PPF ratio is used to discard poor or noisy transitions in our hybrid-DIA pipeline (see Supplementary Data 2 for more details). In gray, number of heavy-endogenous pairs with "dot product ratio" or DOTPR bigger or equal to 0.5. DOTPR measures whether the transition peak areas in the two label types are in the same ratio to each other, and it allows to discard transition that do not match

the expected fragmentation profile. Finally, In red, number of heavy-endogenous pairs that can be quantified with at least 3 fragments (criteria used for hybrid-DIA quantification analysis in this experiment).

Overall, these new analysis shows that the depth and sensitivity of the targeted peptides is higher when using hybrid-DIA MSx scans than standard DIA analysis.

Rebuttal Letter Figure 7:

Next, we focused on the quality of the quantification of the targeted peptides by means of accuracy and precision. First, we have included a correlation analysis plot of quantified heavy-endogenous ratios between replicates and input amounts for DIA (left) and hybrid-DIA (right) analysis (panel C above in rebuttal figure 7, and revised Figure 3E). For DIA analysis, we used the endogenous-heavy peptide pairs that were quantified in Skyline with a ppfr > 0.5. For hybrid-DIA analysis, we used the endogenous-heavy peptide pairs quantified using our shiny-app (ppfr > 0.5 and 3 valid fragments). From this analysis, we can conclude that the correlation between replicates is better when measuring the targeted peptides using hybrid-DIA MSx scans.

Finally, we evaluated the accuracy of the measurements by comparing the ratio distribution between different starting input amounts (panel D from rebuttal letter 7, and revised Figure 3E). Since there are many variables in phospho-peptide enrichment (such ratio peptide-beads, peptide-affinity and peptide-beads stoichiometry, which were not scaled for the input amount in this experiment), we do not expect linearity when measuring the phosphoproteome obtained from 30 µg of peptide against that measured from 20, 10 5 and 2.5 µg. However, the ratio from two replicates of 30 µg of input material is expected to be 0 (in log₂ scale) and it must increase when we divided the sample with 30 µg of input material to the others of decreasing amount. Next, we can use the distribution of ratios measured from the full phospho-proteome as a reference of the expected values, and compare it to the values obtained from the endogenous-heavy pairs of the monitored/targeted peptides.

We have represented this analysis using boxplots (panel D from rebuttal letter 7, and revised Figure 3E). Boxplots show the phospho-site ratio distribution between different conditions in DIA (left) and hybrid-DIA (right) analysis. Each plot shows the ratios from two replicates of 30 µg of input material, 30 µg vs 20 µg, 30 µg vs 10 µg, 30 µg vs 5 µg and 30 µg vs 2.5 µg. For each comparison, there are two boxplots, one in dark and one in light color. The first one (dark) shows the global distribution of all phospho-sites measured using the DIA windows in both methods and analyzed in Spectronaut (i.e: global phosphoproteome). The second one (light) shows the distribution of the endogenous/heavy ratios of the monitored/targeted phospho-peptides. Whilst the ratios for the full phospho-proteome (dark green and dark blue) boxplots are equivalent in DIA and hybrid-DIA, it seems clear that the accuracy when looking at the endogenous-heavy peptides decreases significantly in DIA when the ratios becomes bigger, while in hybrid-DIA the distribution of the endogenous-heavy ratios is on line with the values obtained from the full phosphoproteome.

This systematic comparison of DIA and hybrid-DIA for phosphoproteomics has been included in the revised manuscript (lines 254-300) and in the new Figure 3.

Overall, we expect this new analysis would address the concerns of this reviewer and help clarify the benefits of using hybrid-DIA versus DIA, in terms of sensitivity, precision and accuracy, when monitoring a pre-defined set of peptides using spiked-in heavy labelled standards.

Fig2E shows lower correlation between DIA and hybrid-DIA when the sample injection amount decreases, but which is closer to the ground truth? Reader might expect hybrid-DIA is more quantitatively precise for some peptides with targeted analysis, but how about the others? The authors should largely expand the analysis of this section so that readers could obtain a more comprehensive view of the benefits (and limitations, if any) of hybrid-DIA.

We agree with the observation made by this reviewer. As we have demonstrated before, hybrid-DIA is more quantitatively precise for the phosphor-peptides that are part of the targeted analysis. On the other hand, when looking at the other peptides (full phosphor-proteome), which are quantified from the DIA windows, we observed some discrepancies between doing hybrid-DIA and doing standard DIA acquisition. We initially reported that the depth of the phospho-proteome is slightly affected when using hybrid-DIA, and that the number of identified phospho-sites decreases with increasing number of targeted peptides contained in the inclusion list. We provided an analysis of this effect as part of the revised Figure 4 (and as Rebuttal Letter Figure 8, below). Here we showed that increasing the number of targets consumes a higher proportion of the duty cycle (panel A) and have an impact on the identified phosphor-sites (panel B). Due to this reason, we advise in the manuscript to choose the targeted peptides so they are evenly distributed though the gradient, and provide a reference graph (panel C) to predict the expected losses on phospho-proteome identification on hybrid-DIA based on the targets per minute monitored in the hybrid-DIA experiment. This data is reported now in the revised Figure 4.

Rebuttal Letter Figure 8:

On the other hand, as the reviewer pointed out we also must evaluate the quantification precision of the phospho-proteome measured in DIA and hybrid-DIA mode (Revised Figure 4G, and Rebuttal Letter Figure 9). In the panel A below, we plotted the correlation between a hybrid-DIA replicate (y-axis) and a DIA replicate (x-axis) of different starting amounts for phospho-enrichment. As the reviewer pointed out, the correlation decreased with decreasing input amount. This observation is expected, and in fact, if we plot the correlation between replicates of different input amounts of the DIA analysis (panel B); we can see that the correlation also tends to decrease with lower input amounts.

Rebuttal Letter Figure 9:

With this, we expect to show that the decrease in correlation with lower input amount is not intrinsic to hybrid-DIA analysis, but it is an overall trend observed in quantitative proteomics. Quantification precision is inversely proportional to MS intensity, so lower input amounts that would give lower MS intensity are expected to be less precise, in all kind of MS-based quantification techniques. In addition, the number of phospho-site decreases with lower input amounts, and lower number of points also can have an impact on the calculated correlation. This fact has already been addressed in a previous publication from our group (Koenig et al, Proteomics, 2022 PMID: 35713889) where we clearly show

the impact on precision (and therefore replicate correlation) when decreasing the peptide-input amount (Supplementary Figure 6, from Koenig et al, and adapted as Rebuttal Letter Figure 10, below).

Rebuttal Letter Figure 10 (adapted from Koenig et al, Proteomics 2022)

In Fig3, the authors showed hybrid-DIA generated more data than SureQuant. Again, the authors did not show systematically and quantitatively how much % of accuracy was improved or sacrificed using hybrid-DIA compared to SureQuant. They showed some profiles of selected phosphopeptides based on a signaling pathway, however, for this methodology paper, more analytical analyses are recommended.

We apologized if our statements in the text were not clear. We did not presume that the benefits of hybrid-DIA when comparing it to SureQuant is due to better accuracy in the quantification of the targeted peptides. On the contrary, we use SureQuant as a reference to assess the quality of the hybrid-DIA MSx quantification. Our claim about the benefits of hybrid-DIA versus SureQuant relies on the extra layer of information obtained when using hybrid-DIA, since we obtained, not only the targeted peptides, but the general phosphoproteome from the DIA scans. This extra layer of information is what allowed us to perform kinase activity inference in the EGF stimulation experiment.

However, to reinforce the benchmark of the targeted quantification in hybrid-DIA against SureQuant, we have included a new analysis, where we plot the ratio in the A549 dilution series (Rebuttal Letter Figure 11, panel A) experiment of 30 µg of input material against the other input amounts. As expected, and to confirm that hybrid-DIA provide the same level of accuracy and precision as SureQuant, the boxplot show similar median and distribution for both methodologies (Rebuttal Letter Figure 11, panel B).

In summary, we have included this new analysis in the revised manuscript (lines 364-370 and Figure 4B). Also, we have rewritten the text to make more clear the benefits and disadvantages of each method (lines 373-385). Besides, for this purpose, we have included a comparative table of all the methods assessed in this manuscript (Figure 5C, Rebuttal Letter Figure 11, panel C) to point out their benefits and weaknesses.

Rebuttal Letter Figure 11

Minor issues:

1. It would be better to briefly clarify the importance of comprehensively quantification of phosphoproteomics in biological or clinical samples in the introduction.

We have rewritten the last section of the introduction (lines 49-60) to highlight the importance of depth in phosphoproteomics quantitative analysis, especially in clinical samples or other set-ups where the sample is limited.

2. Background for library free phosphoproteomics on page 6 should be put in the introduction.

We have edited the introduction to include a background on library free phosphoproteomics analysis (lines 39-41).

3. Please add more legends or text sign in the Figure 1A to help understand the acquisition strategy, e.g., the red and purple rectangles for PRM and MSx, respectively. Besides, if the DIA data are simultaneously acquired with PRM and MSx as described on page 5, the green rectangles standing for DIA windows should not be put behind the red and purple rectangles denoting PRM and MSx.

We thanks the reviewer for this suggestion to make the method more clear and understandable. First, we have edited the legend as suggested by this reviewer to include a detailed description of the content.

Secondly, regarding the simultaneous acquisition that this reviewer refer to, we apologize for our choice of "simultaneous" as a word to indicate how the MSx scans are acquired. We said "simultaneous" to indicate that the MSx and DIA acquisition was obtained in the same MS run, but not at the same time. This method works in an Orbitrap Exploris 480, so we only have one detector, therefore not allowing to perform "online" acquisition. The MSx scans are acquired sequentially and added in the duty cycle with the rest of the DIA windows, in the corresponding cycle where the heavy peptide has previously detected. We have corrected the text, to make this clear (legend Figure 1A, lines 128-143, lines 571-573).

Also, we have updated the title to: "Hybrid-DIA: Intelligent Data Acquisition for Online Targeted and Discovery Proteomics Applied to Phosphoprotein signaling in Single Spheroids" to avoid confusion due to the use of the term "simultaneous".

4.The reviewer advises changing Figure 2D from the current line plot to histogram, and labelling the phosphor-site identification numbers for clarity.

We have re-made the figure to be in barplot format. We included the numbers in the source-data file, due to restriction in the size on the figures.

5.The reviewer advises adding some statistical test in Figure 3C.

We have taken into account the suggestion, but In order to keep the figure not too crowded, we have provided the statistical information from that figure as a supplementary table (Supplementary Table S1A). We also added a new panel (Figure 6A) with the list of significantly regulated sites (q-value <0.05, log₂ FC >1.5).

6.What does the activity in Figure 3D mean and how was it assessed?

The activity on the figure is obtained from ROKAI algorithm described in Yilmaz et al (PMID: 33608514). ROKAI performs an inference of kinase activity using functional networks obtained from integrating various sources of functional information: PhosphoSitePlus and Signor for kinase substrate annotation, and String and PTMcode, among others, for protein-protein interaction, structure distance and co-evolution of kinases analysis. ROKAI then integrates this information with the phosphorylation profile (provided as a ranked list, in our case, the log₂ fold change ratios in each time point versus time zero). As a result, ROKAI predict the kinase activities and calculate the statistical significance of such predicted activity. For that, the algorithm employs a weighted Z-test based on the phosphorylation of known kinase targets and other phosphosites in their functional neighborhood (since RoKAI utilizes a linear model, the inferred activity of a kinase can be decomposed as a weighted summation of site phosphorylations, thus, the statistical significance can be computed using a weighted z-test). During this test, the standard deviation across all sites is used to estimate the standard error. All this information is available in the ROKAI original publication. To facilitate understanding what do we refer for activity in this figure, we have change the name to “Inferred Kinase activity” and added the reference to the ROKAI publication into the legend, as well as a better description of the plot in the legend.

There is no annotation for the yellow group in Figure 3E or its legend. I guess yellow is for the group without any inhibitor? Please clarify.

We are sorry we did not specify this in the legend. The reviewer is right, the yellow box correspond to the data with EGF stimulation without any kinase inhibitor. We have modified the figure to indicate it, as well as the figure legend.

7.Please specify in Figure 4C which panel belongs the results of spheroid or monolayer. It is better to display the figure in the form of split violin plot.

We have corrected the figure to add the information on which panel belong to 2D or 3D conditions. We have considered the suggestion of changing the plot into a split violin plot. However, in this case the intensities of 2D are overall higher than those of 3D, due to higher loading into the MS instrument because of higher peptide recovery. Therefore, the boxplots for 2D and 3D won't be aligned if we plot them as a split violin plot.

8. There are several typos and inappropriate expressions. Some examples were listed as following:

Page 3: Change "required" to "require".

Figure 4B: Change "Spheroids" to "Spheroid".

Thanks for pointing out these typos, we have corrected them.

REVIEWERS' COMMENTS

Reviewer #1 (Remarks to the Author):

Ana Martínez-Val et al. have done a very thorough job in this revision. I am fully satisfied with their answers to my questions and in my opinion, they also address the other two reviewer's questions more than adequately. Style and presentation have also improved although the authors should do some more proofreading.

Reviewer #2 (Remarks to the Author):

I am writing to inform that I have received satisfactory answers to my questions from the authors. The explanations provided have addressed my concerns, and I have no further suggestions for changes or additional experiments.

Additionally, I would like to express my support for the acceptance of the manuscript. I believe the method presented will be beneficial to the scientific community and contribute to advancing our understanding in the field.

Best regards

Mårten

Reviewer #3 (Remarks to the Author):

Thanks for the revision. I have no more comment.

RESPONSE TO REVIEWERS' COMMENTS

We appreciate the positive feedback of all reviewers and we would like to thank them for their suggestions during the peer-review process which have improved the quality of the work.

Reviewer #1 (Remarks to the Author):

Ana Martínez-Val et al. have done a very thorough job in this revision. I am fully satisfied with their answers to my questions and in my opinion, they also address the other two reviewer's questions more than adequately. Style and presentation have also improved although the authors should do some more proofreading.

Following the reviewer's suggestion, we have done more proofreading to correct typos and improve sentence coherence in the text.

Reviewer #2 (Remarks to the Author):

I am writing to inform that I have received satisfactory answers to my questions from the authors. The explanations provided have addressed my concerns, and I have no further suggestions for changes or additional experiments.

Additionally, I would like to express my support for the acceptance of the manuscript. I believe the method presented will be beneficial to the scientific community and contribute to advancing our understanding in the field.

We would like to thank the reviewer for their positive feedback.

Reviewer #3 (Remarks to the Author):

Thanks for the revision. I have no more comment.

We are glad that our revised manuscript answered the original concerns by the reviewer.